# Air pollution satellite-based CO₂ emission inversion: system evaluation, sensitivity analysis, and future research direction

Hui Li[1,2], Jiaxin Qiu[1,2], Bo Zheng[1,2,]*

[1]Shenzhen Key Laboratory of Ecological Remediation and Carbon Sequestration, Institute of Environment and Ecology, Tsinghua Shenzhen International Graduate School, Tsinghua University, Shenzhen 518055, China.

[2]State Environmental Protection Key Laboratory of Sources and Control of Air Pollution Complex, Beijing 100084, China.

*Correspondence to*: Bo Zheng (bozheng@sz.tsinghua.edu.cn)

**Abstract.** Simultaneous monitoring of greenhouse gases and air pollutant emissions is crucial for combating global warming and air pollution. We previously established an air pollution satellite-based carbon dioxide ($CO_2$) emission inversion system, successfully capturing $CO_2$ and nitrogen oxides ($NO_x$) emission fluctuations amid socioeconomic changes. However, the system's robustness and weaknesses have not yet been fully evaluated. Here, we conduct a comprehensive sensitivity analysis with 31 tests on various factors including prior, model resolution, satellite constraint, and inversion system configuration to assess the vulnerability of emission estimates across temporal, sectoral, and spatial dimensions. The Relative Change (*RC*) between these tests and Base inversion reflects the different configurations' impact on inferred emissions, with one standard deviation ($1\sigma$) of *RC* indicating consistency. Although estimates show increased sensitivity to tested factors at finer scales, the system demonstrates notable robustness, especially for annual national total $NO_x$ and $CO_2$ emissions across most tests (*RC* < 4.0%). Spatiotemporally diverse changes in parameters tend to yield inconsistent impacts ($1\sigma \geq 4\%$) on estimates, and vice versa ($1\sigma < 4\%$). The model resolution, satellite constraint, and $NO_x$ emission factors emerge as the major influential factors, underscoring their priority for further optimization. Taking daily national total $CO_2$ emissions as an example, the $\overline{RC} \pm 1\sigma$ they incur can reach -1.2±6.0%, 1.3±3.9%, and 10.7±0.7%, respectively. This study reveals the robustness and areas for improvement in our air pollution satellite-based $CO_2$ emission inversion system, offering opportunities to enhance the reliability of $CO_2$ emission monitoring in the future.

## 1 Introduction

The knowledge of emissions, i.e., how much, where, and by what activity pollutants are released into the atmosphere, lays the foundation for understanding the changes in atmospheric compositions and managing emissions toward climate and air quality targets (Meinshausen et al., 2022; Li et al., 2022; Zhang et al., 2019). Anthropogenic emissions are strongly modulated by socioeconomic events (e.g., holidays, economic recession, and recovery), therefore, it is essential to monitor emissions timely to interpret atmospheric species

concentrations (Shan et al., 2021; Le Quéré et al., 2021; Guevara et al., 2023). Currently, numerous nations,
particularly those within the Global South (i.e., China), grapple with the dual imperatives of mitigating air
pollution and addressing climate change challenges. To effectively navigate these intertwined challenges in
a harmonized and resource-efficient manner, the development of a system capable of disentangling variations
in emissions and their driving factors for greenhouse gases and air pollutants is indispensable (Ke et al., 2023).
Recently, a discernible trend is emerging towards inferring anthropogenic carbon dioxide ($CO_2$) emissions
from well-observed and co-emitted air pollutants (i.e., nitrogen dioxide, $NO_2$) given their co-emission
characteristics in time and space (Wren et al., 2023; Yang et al., 2023; Liu et al., 2020a; Reuter et al., 2019).
$NO_2$ forms rapidly after NO is emitted from sources and is also the primary nitrogen oxide detectable by most
satellites (Ye et al., 2016). This makes $NO_2$ a reliable and widely adopted proxy in nitrogen oxides ($NO_x$ =
$NO_2$+NO) emission inversions. However, the co-emission of $NO_x$ and $CO_2$ does not imply synchronized
trends in their emissions, as the $CO_2$-to-$NO_x$ emission ratios and activity trends vary across different sectors
(Li and Zheng, 2024). The introduction of $NO_2$ in the $CO_2$ emission estimation presents several distinct
advantages. $NO_2$ has a short lifetime of several hours, rendering its source-contributing plumes readily
detectable via remote sensing techniques (Goldberg et al., 2019). This short lifespan of $NO_2$ facilitates mass-
balance approaches for estimating $NO_x$ emissions, which rely on the assumption of a linear relationship
between $NO_2$ columns and local $NO_x$ emissions (Cooper et al., 2017; Mun et al., 2023; Martin et al., 2003).
In contrast, the longevity of $CO_2$, spanning hundreds of years, combined with its elevated background
concentration reaching hundreds of parts per million (ppm), obscures the detection of local source-triggered
concentration enhancements (i.e., several ppm) (Nassar et al., 2017; Reuter et al., 2019). Moreover, remote
sensing technologies for $NO_2$ remain generally more mature, as indicated by the broader coverage and
improved signal-to-noise ratio in column concentration observation (Macdonald et al., 2023; Cooper et al.,
2022). Recent advancements in $CO_2$ satellite technology are promising, such as the Orbiting Carbon
Observatory-3 (OCO-3), which can generate $CO_2$ maps with a resolution of up to 1.6 km × 2.2 km and
monitor $CO_2$ columns at different times throughout the daytime to elucidate diurnal emission patterns (Taylor
et al., 2023), while its spatial coverage may not be sufficient for large-area inversions at high temporal
resolution. The synergistic quantification of $CO_2$ and $NO_x$ emissions has gained substantial attention, not to
mention that it could provide valuable guidance for a joint effort to monitor and mitigate air pollutants and
carbon emissions concurrently (Miyazaki and Bowman, 2023).
We have developed an air pollution satellite sensor-based $CO_2$ emission inversion system, which is capable
of concurrently estimating the ten-day moving average of sector-specific anthropogenic $NO_x$ and $CO_2$
emissions by integrating top-down and bottom-up methods. This integrated methodology has proven
effective in capturing emission fluctuations, particularly during the coronavirus disease 2019 (COVID-19)
pandemic (Zheng et al., 2020; Li et al., 2023). While previous sensitivity tests have suggested a certain level
of accuracy, the system has not yet undergone a comprehensive evaluation to thoroughly assess its robustness
and weaknesses, and thereby clearly imply its future developmental trajectory. To bridge this gap, we
undertake an extensive sensitivity analysis with 31 tests using the 2022 anthropogenic $NO_x$ and $CO_2$ emission
estimation as a case study. This study investigates how emission outcomes respond to a variety of sensitivity
assessments across temporal, sectoral, and spatial dimensions. This study aims to diagnose and rank the
uncertainty sources, providing insights to prioritize improvements of this inversion system in the future.
**2 Materials and methods**
Our air pollution satellite sensor-based $CO_2$ emission inversion system has been elucidated in our previous
studies (Zheng et al., 2020; Li et al., 2023). In essence, this system integrates top-down and bottom-up data
streams to infer the ten-day moving average of anthropogenic $NO_x$ and $CO_2$ emissions by sector in China
based on the mass-balance approach (Cooper et al., 2017). Comprising three key components, the system
involves the bottom-up inference of prior emissions for $NO_x$ and $CO_2$ with sectoral profile, the top-down
estimation of total $NO_x$ emissions constrained by satellite observation, and the integration of both sources to
derive satellite-constrained $NO_x$ and $CO_2$ emissions by sector (Fig. S1). Each of these processes could
introduce uncertainties in the final emission estimates. To assess the potential uncertainties, we establish a
baseline (Base) for emissions computed using our conventional settings (Li et al., 2023; Zheng et al., 2020)
and further investigate sensitivity tests to characterize the impacts of the different configurations on final
estimates.
**2.1 Inversion methodology and Base inversion**
We use the Base inversion as a case to provide a detailed explanation of this inversion system. In the Base
inversion, we adhered to the same parameters and configurations outlined in previous studies for estimating
the ten-day moving average of anthropogenic $NO_x$ and $CO_2$ emissions by sector in 2022 (Table 1) (Li et al.,
2023; Zheng et al., 2020). Succinctly, we first updated sectoral $NO_x$ and $CO_2$ emissions from the Multi-
resolution Emission Inventory for China (MEIC) inventory (Zheng et al., 2018) through the bottom-up
process. This involved utilizing indicators including industrial production, thermal power generation, freight
turnover, and population-weighted heating degree days as proxies for changes in industry, power, transport,
and residential activity levels (Details seen in Text S1 and Table S1). Notably, to reconcile the resolution
between the prior emissions and the model, we aggregated the original MEIC emissions from a resolution of
$0.25° \times 0.25°$ (Fig. S2) to $0.5° \times 0.625°$. Secondly, we inferred the total anthropogenic $NO_x$ emissions
constrained by TROPOspheric Monitoring Instrument (TROPOMI) $NO_2$ retrievals (v2.4) (Van Geffen et al.,
2022) (Eq. 1). A critical step in this process was establishing a linear relationship between $NO_2$ tropospheric
vertical column densities (TVCDs) and anthropogenic $NO_x$ emissions under the mass balance assumption
(Eq. 2) through GEOS-Chem simulation (v12.3.0, https://geoschem.github.io/) at a horizontal resolution of
$0.5° \times 0.625°$. Our analysis focused on the grids where anthropogenic emissions prevail (Liu et al., 2020b),
characterized by a ten-day moving average of $NO_2$ TVCDs exceeding $1 \times 10^{15}$ molecules cm$^{-2}$.
$$E_{t,i,TROPOMI,y} = (1 + \beta_{t,i} (\frac{\Delta\Omega}{\Omega})_{t,i,anth,y}) \times E_{t,i,bottom-up,2019} \qquad (1)$$

$$\beta_{t,i} = \frac{\Delta E_{t,i,bottom-up,2019}}{E_{t,i,bottom-up,2019}} \div \frac{\Omega_{t,i,-40\%emi,2019} - \Omega_{t,i,\text{base},2019}}{\Omega_{t,i,\text{base},2019}} \qquad (2)$$
$$(\frac{\Delta\Omega}{\Omega})_{t,i,anth,y} = \frac{\Omega_{t,i,sate,y}}{\Omega_{t,i,sate,2019}} - \frac{\Omega_{t,i,simu\_fixemis,y}}{\Omega_{t,i,simu,2019}} \qquad (3)$$
Where $t$, $i$, and $y$ represent the ten-day window, model grid cell (i.e., 0.5°×0.625°), and target year 2022,
respectively. $E_{t,i,\text{TROPOMI},y}$ is the anthropogenic total $NO_x$ emissions constrained by TROPOMI $NO_2$ TVCDs.
$E_{t,i,\text{bottom-up},2019}$ is the anthropogenic $NO_x$ emissions in 2019 from the MEIC. $\beta_{t,i}$ is a unitless factor relating the
changes in $NO_2$ TVCDs to anthropogenic $NO_x$ emissions (Lamsal et al., 2011). $\Delta E_{t,i,\text{bottom-up},2019}/E_{t,i,\text{bottom-up},2019}$
represent the implemented 40% reduction in anthropogenic $NO_x$ emissions over China. The 40% reduction
was selected after a series of sensitivity tests, which demonstrated that this perturbation level exerts a limited
impact on the $\beta$ estimates (Zheng et al., 2020). $\Omega_{t,i,-40\%emi,2019}$ and $\Omega_{t,i,\text{base},2019}$ are GEOS-Chem simulated $NO_2$
TVCDs at the TROPOMI overpass time in 2019 with a 40% emission reduction and without any emission
reduction, respectively. $(\Delta\Omega/\Omega)_{t,i,anth,y}$ refers to the relative changes in $NO_2$ TVCDs due to anthropogenic $NO_x$
emission changes between 2019 and 2022. $\Omega_{t,i,sate,y}/\Omega_{t,i,sate,2019}$ indicates the relative differences in TROPOMI
$NO_2$ TVCDs between 2019 and 2022, and $\Omega_{t,i,simu\_fixemis,y}/\Omega_{t,i,simu,2019}$ represents the relative changes in $NO_2$
TVCDs caused by inter-annual meteorological variation, which are derived from GEOS-Chem simulations
with the fixed 2019 emissions and meteorological field in target year.
Thirdly, we integrated the bottom-up and top-down data flows to yield TROPOMI-constrained sectoral $NO_x$
emissions. Assuming that each grid's emission variability was primarily driven by its dominant source sectors
(contributing over 50%), we utilized the discrepancy between the bottom-up and top-down estimates in grid
cells dominated by a particular sector to derive sector-specific scaling factors, which were subsequently
applied to correct the bottom-up sectoral $NO_x$ emissions (Eq. 4). For grids without a sector contributing over
50%, we excluded them from sectoral scaling factor calculations, instead applying scaling factors derived
from grids meeting this criterion. The number of these grids accounts for less than 20% of total grids, making
their impact negligible. Following this adjustment, we rescaled the corrected bottom-up emissions to ensure
alignment with the TROPOMI-constrained total emissions. The overall sectoral correction factors mainly
range from 0.5 to 1.5 (Fig. S3).
$$\text{scalefactor}_{t,s,y} = 1 + \frac{\sum_i (E^s_{t,i,\text{sate},y} - E^s_{t,i,\text{bottom-up},y})}{\sum_i E^s_{t,i,\text{bottom-up},y}} \qquad (4)$$
Where $t$, $s$, $i$, and $y$ represent the ten-day window, sector, grid cell (i.e., 0.5°×0.625°), and year 2022,
respectively. $E^s_{t,i,\text{sate},y}$ and $E^s_{t,i,\text{bottom-up},y}$ are TROPOMI-constrained and bottom-up estimated $NO_x$ emissions
on grid cell $i$ with dominated source sector $s$, respectively. The scalefactor$_{t,s,y}$ is the scaling factor used to
correct the bottom-up estimated $NO_x$ emissions from sectors in time $t$ in year $y$.
Finally, we converted the sectoral $NO_x$ emissions to corresponding $CO_2$ emissions with the $CO_2$-to-$NO_x$
emission ratios derived from the bottom-up process (Eq. 5). The $CO_2$-to-$NO_x$ emission ratios in 2022 are
updated by reducing $NO_x$ emission factors (EFs) while keeping $CO_2$ EFs unchanged based on 2019 MEIC.
The default assumption that the reduction rate halves annually is due to the limited potential for further
reductions. In contrast, the $CO_2$ EFs are assumed to remain unchanged, as they are primarily determined by
fuel type and combustion conditions (Cheng et al., 2021) (details seen in Text S2).
$$C_{s,t,i,TROPOMI,y} = E_{s,t,i,TROPOMI,y} \times \frac{EF_{CO_2\,s,i,bottom-up,2019}}{EF_{NO_x\,s,i,bottom-up,2019} \times (1 - rNO_{x\,s,i,y})}$$
(5)

Where $C_{s,t,i,\text{TROPOMI},y}$ and $E_{s,t,i,\text{TROPOMI},y}$ are $CO_2$ and $NO_x$ emissions from sector $s$. $EF_{co_2\,s,i,\text{bottom-up},2019}$ and
$EF_{NOx\,s,i,\text{bottom-up},2019}$ are the sectoral EFs of $CO_2$ and $NO_x$ in 2019 derived from the MEIC emission model.
$rNO_{x\,s,i,y}$ is the reduction ratio in $NO_x$ EFs by sector from 2019 to 2022 derived from the bottom-up estimation.
We approximate the annual $NO_x$ and $CO_2$ emissions as the sum of the ten-day moving average of $NO_x$ and
$CO_2$ emissions in 2022 with a vacancy in the first and last five days. This approximation, however, does not
impact our analysis, as our primary objective is to identify potential sources of uncertainty within the system
and thereby highlight areas for future improvement.
**Table 1. Configurations of Base inversion.**

| Factors/parameters | Base setting |
| --- | --- |
| GEOS-Chem (GC) resolution | GEOS-Chem simulation with the resolution of 0.5°×0.625° |
| TROPOMI retrievals version | v2.4 of TROPOMI $NO_2$ |
| TROPOMI screening schemes | Cloud fraction (CF)<0.4, quality flag (QA)>0.5 |
| Reference year | 2019 |
| $NO_x$ emission factors (EFs) | The reduction ratio of $NO_x$ EFs halves annually* |
| Threshold value to identify dominant emission source sectors for each grid | 50% |
| Sectors in bottom-up estimation | 8 sectors (power, industry, cement, iron, residential, residential-bio, on-road, and off-road) |

*Each year's reduction rate for $NO_x$ EFs is set to decrease by half compared to the previous year. For example, if the reduction of $NO_x$
EFs from 2019 to 2020 was 4%, the reduction from 2020 to 2021 would be set at 2%.
**2.2 Sensitivity settings**
The sensitivity inversion experiments comprise 31 tests designed to provide a comprehensive evaluation of
the system. To facilitate a clearer discussion of their impacts, we categorized these tests into four classes
based on their roles within the system: prior information, GEOS-Chem model resolution, satellite
observational constraints, and inversion system parameters (Fig. 1 and Table 2). Each test is conducted as a
controlled experiment, where only one parameter is altered while the rest remain the same as their Base
inversion setting. The rationale behind the settings and their design will be elaborated in the following
sections.
**Table 2. Settings of 31 sensitivity inversion tests.**

| Category | Num | Name | Settings description | Test objectives |
|---|---|---|---|---|
| GC | 1 | Res_2×2.5 | GEOS-Chem simulation with the resolution of 2°×2.5° | Model resolution |
| Satellite constraint | 2 | Trop_fill | Complementing TROPOMI $NO_2$ with machine learning | Sampling coverage |
| | 3 | Trop_v2.3 | Substituting TROPOMI $NO_2$ from v2.4 to v2.3 | Satellite data version |
| | 4 | Trop_cf03 | Changing CF limit from 0.4 to 0.3 | Satellite data filtering condition |
| | 5 | Trop_cf05 | Changing CF limit from 0.4 to 0.5 | |
| | 6 | Trop_qa06 | Changing QA limit from 0.5 to 0.6 | |
| | 7 | Trop_qa07 | Changing QA limit from 0.5 to 0.7 | |
| Inversion system parameters | 8 | 2021_base | Changing the reference year from 2019 to 2021 | Reference year |
| | 9 | β_-20% | Scaling $\beta$ down by 20% | $\beta$ |
| | 10 | β_-15% | Scaling $\beta$ down by 15% | |
| | 11 | β_-10% | Scaling $\beta$ down by 10% | |
| | 12 | β_-5% | Scaling $\beta$ down by 5% | |
| | 13 | β_-1% | Scaling $\beta$ down by 1% | |
| | 14 | β_1% | Scaling $\beta$ up by 1% | |
| | 15 | β_5% | Scaling $\beta$ up by 5% | |
| | 16 | β_10% | Scaling $\beta$ up by 10% | |
| | 17 | β_15% | Scaling $\beta$ up by 15% | |
| | 18 | β_20% | Scaling $\beta$ up by 20% | |
| Prior | 19 | ef_-10% | Scaling changes in $NO_x$ EFs down by 10% | $NO_x$ EFs |
| | 20 | ef_-9% | Scaling changes in $NO_x$ EFs down by 9% | |
| | 21 | ef_-8% | Scaling changes in $NO_x$ EFs down by 8% | |
| | 22 | ef_-7% | Scaling changes in $NO_x$ EFs down by 7% | |
| | 23 | ef_-6% | Scaling changes in $NO_x$ EFs down by 6% | |
| | 24 | ef_-5% | Scaling changes in $NO_x$ EFs down by 5% | |
| | 25 | ef_-4% | Scaling changes in $NO_x$ EFs down by 4% | |
| | 26 | ef_-3% | Scaling changes in $NO_x$ EFs down by 3% | |
| | 27 | ef_-2% | Scaling changes in $NO_x$ EFs down by 2% | |
| | 28 | ef_-1% | Scaling changes in $NO_x$ EFs down by 1% | |
| | 29 | thre_40% | Changing the dominant sector threshold from 50% to 40% | Threshold |
| | 30 | thre_60% | Changing the dominant sector threshold from 50% to 60% | |
| | 31 | 4_sectors | Aggregating the sectors from 8 to 4 in prior estimates | Sector's classification |


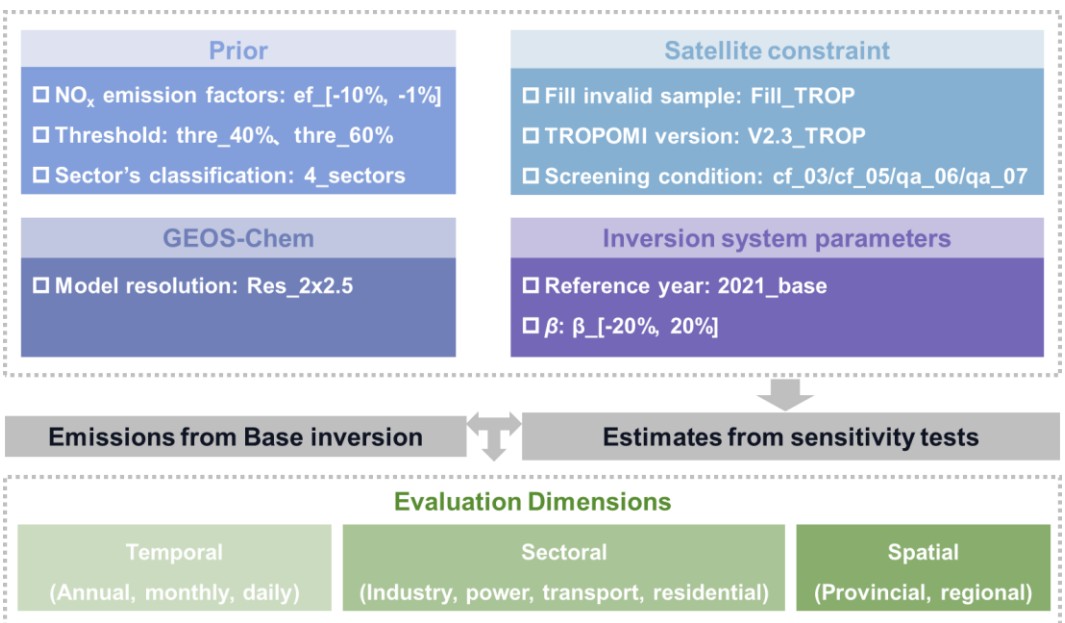

Figure 1. **Overview of the sensitivity inversion tests in this study.** Details of the processes and settings are presented in Fig. S1 and Table 2.

**2.2.1 Modifying prior emission estimates**

The prior provides the sectoral profile for subsequent emission attribution. We conducted a comprehensive examination of associated parameters when updating the prior from 2019 MEIC ($0.5° \times 0.625°$), including $NO_x$ EFs influencing the conversion of $NO_x$ to $CO_2$ emissions by sector, threshold value defining the dominant sector for each grid, and sector classification. For $NO_x$ EFs settings, we devised a ten-level gradient ranging from -10% to -1% (referred to as ef_[-10%, -1%]). Regarding the threshold value, we varied it from 50% to 40% and 60% (referred to as thre_40% and thre_60%), respectively. For sector classification, the original prior $NO_x$ and $CO_2$ emissions were updated based on eight sectors in the bottom-up process: power, industry, cement, iron, residential, residential-bio, on-road, and off-road. This detailed sectoral structure facilitates relatively detailed bottom-up estimations with specific sectoral activity levels. These eight sectors were then aggregated into four categories: power, industry (sum of original industry, cement, and iron), residential (sum of original residential and residential-bio), and transport (sum of original on-road and off-road) when allocating TROPOMI-constrained total $NO_x$ emissions into sectors. Here, this sector consolidation, specifically implemented before the bottom-up estimation (4_sectors), was designed to evaluate the influence of sector classification on the inversion results.

**2.2.2 Employing coarser model resolution**

The model resolution of the GEOS-Chem simulation inherently shapes the localized relationship between $NO_2$ TVCDs and $NO_x$ emissions established in the top-down process. Finer resolution is advantageous for establishing localized connections between air pollutant emissions and atmospheric concentrations, and the attribution of sectoral emissions. However, excessively fine resolution is not applicable due to the inter-grid

transport when employing the mass-balance method (Turner et al., 2012). To explore the impact of resolution on emission estimates, we performed an inversion experiment with simulations at a coarser resolution of $2° \times 2.5°$ (Res_2×2.5).

**2.2.3 Changing satellite observational constraints**

The TROPOMI $NO_2$ retrievals serve as a constraint in the top-down $NO_x$ emission estimation. We conducted experiments on the TROPOMI $NO_2$ retrievals through three distinct approaches. Firstly, we used Extreme Gradient Boosting (XGBoost) to fill the invalid satellite retrievals in v2.4 TROPOMI (Trop_fill) by establishing relationships between TROPOMI $NO_2$ TVCDs and meteorological variables, as well as GEOS-Chem simulated $NO_2$ TVCDs (modeled_$NO_2$ in Eq. 6) (Wei et al., 2022). The meteorological variables were derived from European Centre for Medium-Range Weather Forecasts (ECMWF) ERA5 dataset (Hersbach et al., 2020), including boundary layer height (BLH), surface pressure (SP), temperature (TEM), dewpoint temperature (DT), 10m u-component (WU), 10m v-component of winds (WV), total precipitation (TP), evaporation (EP), downward uv radiation at the surface (surUV), and mean surface downward uv radiation flux (downUV). In the XGBoost process, we trained the relationship for daily $NO_2$ TVCDs throughout the year grid-by-grid, with 80% of the data used as the training set and 20% as the test set.

$$TROPOMI\_NO_2 \sim f_{XGBoost}(modeled\_NO_2, BLH, SP, TEM, DT, WU, WV, TP, EP, surUV, downUV) \quad (6)$$

The comparison of $NO_2$ TVCDs before and after data filling revealed minimal impact from the original missing data (Fig. S4). This is attributed to our system's utilization of a ten-day moving average of $NO_2$ TVCDs, which effectively mitigates the influence of missing data at the grid scale.

Secondly, we evaluated the impact of different versions of TROPOMI $NO_2$ retrievals by substituting the v2.4 TROPOMI data with the older v2.3 TROPOMI $NO_2$ columns (Trop_v2.3). Updates in TROPOMI data products generally help address the low bias of $NO_2$ concentrations, particularly in heavily polluted regions (Lange et al., 2023; Van Geffen et al., 2022). Thirdly, we adjusted the satellite data screening protocols to investigate the uncertainties associated with satellite observations on emission estimates, which involved varying the cloud fraction (CF) limit to 0.3 (Trop_cf03) or 0.5 (Trop_cf05) and modifying the quality flag (QA) limit to 0.6 (Trop_qa06) or 0.7 (Trop_qa07), respectively. CF and QA serve as crucial parameters in screening applicable $NO_2$ TVCDs, representing primary sources of uncertainty in satellite observations (Van Geffen et al., 2022; Lange et al., 2023).

**2.2.4 Tests on inversion system parameters**

In previous studies, the reference year for updating emissions for target years was 2019. Here, we modified the reference year to 2021 (2021_base) to assess its impact. The parameter $\beta$ represents the localized relationship between changes in $NO_2$ TVCDs and changes in anthropogenic $NO_x$ emissions (Eq. 2), determining the transition from observed changes in $NO_2$ TVCDs to changes in anthropogenic $NO_x$ emissions

in the top-down process. To explore potential nonlinear responses in the estimated results to this parameter,
we devised a ten-level gradient for $\beta$, ranging from -20% to 20% (refer to as $\beta\_[-20\%, 20\%]$).

## 2.3 Evaluation of different configurations' impact

The sensitivity analysis of the $NO_x$ and $CO_2$ emissions estimated by our inversion system has illuminated
potential sources of uncertainty and the magnitude of their impacts. To quantify the influence of sensitivity
tests on emission estimates, we calculated the Relative Change ($RC$) between emissions estimated under
different tests and the Base inversion, and one standard deviation ($1\sigma$) of $RC$ to evaluate the consistency of
their impact across temporal, sectoral, and spatial scales (details seen in Table 3). It is noteworthy that on the
annual national total emission scale (maximization of all three dimensions), the value of $1\sigma$ equals 0.0%.
**Table 3. Calculation of $RC$ and $1\sigma$ across different dimensions.**

| Dimension | Equations | Parameters |
|---|---|---|
| Temporal | $$RC_t = \frac{E_{t,\text{sensi}} - E_{t,\text{base}}}{E_{t,\text{base}}}$$ $$\sigma_t = \sqrt{\frac{\sum_t^n (RC_t - \overline{RC_t})^2}{n}}$$ | $\cdot$ $t$ represents timescale, denoting year, month, or ten-day window.<br>$\cdot$ $E_{t,\text{sensi}}$ and $E_{t,\text{base}}$ denote the national total emissions under a specific sensitivity test and Base on corresponding temporal scale $t$.<br>$\cdot$ $RC_t$ and $\sigma_t$ indicate the $RC$ and its $1\sigma$ of national total emissions across temporal scales. The $\sigma_t$ equals 0.0% when $t$ is the yearly scale. |
| Sectoral | $$RC_{t,s} = \frac{E_{t,s,\text{sensi}} - E_{t,s,\text{base}}}{E_{t,s,\text{base}}}$$ $$\sigma_s = \sqrt{\frac{\sum_t^n (RC_s - \overline{RC_s})^2}{n}} \quad \text{(Daily)}$$ | $\cdot$ $s$ represents sector source.<br>$\cdot$ $E_{t,s,\text{sensi}}$ and $E_{t,s,\text{base}}$ refer to national sectoral emissions under sensitivity test and Base on temporal scale $t$ (annual and daily).<br>$\cdot$ $RC_{t,s}$ indicates the $RC$ of national sectoral emissions on a temporal scale $t$.<br>$\cdot$ $\sigma_s$ indicates $1\sigma$ of $RC$ of national sectoral emissions on a daily scale. |
| Spatial | $$RC_{t,p/r} = \frac{E_{t,p/r,\text{sensi}} - E_{t,p/r,\text{base}}}{E_{t,p/r,\text{base}}}$$ $$\sigma_p = \sqrt{\frac{\sum_p^m (RC_p - \overline{RC_p})^2}{m}} \quad \text{(Annual)}$$ $$\sigma_r = \sqrt{\frac{\sum_t^n (RC_r - \overline{RC_r})^2}{n}} \quad \text{(Daily)}$$ | $\cdot$ $p$ and $r$ represent province and region (i.e., provincial clusters), respectively.<br>$\cdot$ $E_{t,p/r,\text{sensi}}$ and $E_{t,p/r,\text{base}}$ refer to provincial/regional total emissions under sensitivity test and Base on temporal scale $t$ (annual and daily).<br>$\cdot$ $RC_{t,p/r}$ indicates the $RC$ of provincial/regional total emissions on a temporal scale $t$.<br>$\cdot$ $\sigma_p$ indicates $1\sigma$ of $RC$ of annual total emissions on the provincial scale.<br>$\cdot$ $\sigma_r$ indicates $1\sigma$ of $RC$ of regional total emissions on a daily scale. |


In this context, a condition where $1\sigma$ is below 4.0% is deemed as a consistent impact on emission outcomes
within certain dimensions (the determination of 4.0% seen in Fig. S5). Conversely, when $1\sigma$ exceeds or equals
4.0%, it is indicative of an inconsistent impact. For instance, a daily scale $\sigma_t$ value of 6.2% in the Res_2×2.5
test (Fig. S6) suggests that the model resolution exerts a temporally inconsistent influence on daily emission
estimates, whereas a daily scale $\sigma_t = 0.0\%$ under ef_-10% indicates temporal consistency in its influence.

These principles extend to other dimensions (i.e., sectoral and spatial). Factors whose sensitivity tests yield large and inconsistent $RC$ across finer time, sector, or region scales tend to introduce high uncertainty and become a priority for future optimization. Conversely, small and consistent $RC$ suggests sources with low uncertainty and a higher level of robustness in the system to those particular factors.

## 3 Results

### 3.1 Overview of the emission responses to sensitivity tests

For a comprehensive understanding of emission sensitivity across various dimensions, we compute the sum of absolute average $RC$ and $1\sigma$ (i.e., $\left|\overline{RC}\right|+1\sigma$) to delineate potential most likely uncertainties associated with tested factors across spatial, temporal, and sectoral scales (Fig. 2). The impact of these tests on emissions are comparable between $NO_x$ and $CO_2$, except for the $NO_x$ EFs tests (first column in Fig. 2), which distinctly influence $NO_x$ and $CO_2$ emissions. $CO_2$ emissions display high sensitivity to $NO_x$ EFs across all dimensions compared to $NO_x$ emissions, except in the residential sector where $NO_x$ emissions are more responsive while $CO_2$ emissions are not. For instance, ef_-10% (maximum reduction in $NO_x$ EFs tests) incurs a $\left|\overline{RC}\right|+1\sigma$ of 10.7% in annual national $CO_2$ emissions, with no corresponding impact on $NO_x$ emissions. The relationship between annual national $CO_2$ emissions and $NO_x$ EFs exhibits linearity (Fig. S7), remaining within a 4.0% range if $NO_x$ EFs reductions are kept below 4.0% (i.e., ef_[-4%, -1%]). In contrast, daily residential emissions show a $\left|\overline{RC}\right|$ of only 1.0% in $CO_2$ but up to 9.1% in $NO_x$ emissions under the ef_-10% test.

The remaining sensitivity tests, excluding the $NO_x$ EFs, demonstrate comparable influences on both $NO_x$ and $CO_2$ emissions. Among all dimensions examined, the annual national total $NO_x$ and $CO_2$ emissions emerge as robust results, with a $\left|\overline{RC}\right|+1\sigma$ of no more than 4.0% across tests. At a finer temporal scale (i.e., daily basis), the impacts of model resolution, reference year, and satellite constraint on estimated emissions are amplified, with their $\left|\overline{RC}\right|+1\sigma$ tripling compared to the annual scale. This amplification primarily arises from the increased $1\sigma$ on the daily scale (Fig. S6), indicating the substantial impact of these factors on daily emission estimates. At a finer spatial scale, provincial emissions are vulnerable to changes in model resolution, reference year, and satellite constraint due to their impacts' inconsistency in space (Fig. S6). Concerning sectoral emissions, industry and power sector emissions exhibit robustness, whereas transport and residential emissions present vulnerabilities to model resolution and dominant sector threshold value, respectively. In the following sections, we elaborate on the impacts of all sensitivity tests on $NO_x$ and $CO_2$ emissions from temporal, sectoral, and spatial perspectives. To clarify the $RC$ across different dimensions, we adopt $RC_t$, $RC_s$, and $RC_{p/r}$ to signify $RC$ in temporal, sectoral, and spatial contexts, respectively.

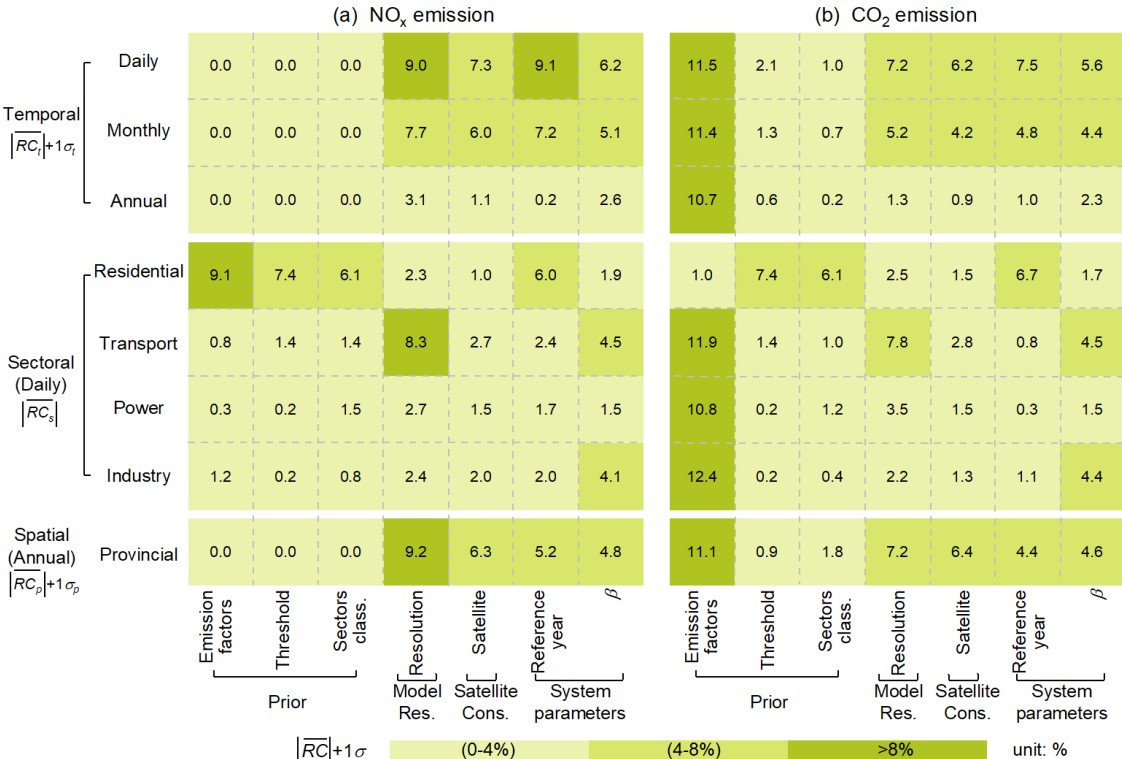

Figure 2. **An overview of sensitivity inversion tests' impacts on (a) NO$_x$ and (b) CO$_2$ emissions**. The color blocks in this figure represent the sum of absolute average $RC$ and $1\sigma$ (i.e., $|\overline{RC}|+1\sigma$), which reflect the extent of the corresponding tests' impact. Sectoral and provincial results are depicted on an annual scale. The numbers within each grid represent the maximum value of $|\overline{RC}|+1\sigma$ under tests on corresponding factors. For example, the $|\overline{RC}|+1\sigma$ noted in the Emission factors column refers to ef_-10%. It is noteworthy that the sectoral dimensions in this figure display their absolute average $RC$ on the daily scale, with their corresponding $1\sigma$ shown separately in Fig. S6.

### 3.2 Emission sensitivity at different temporal scales

To exclusively examine emission sensitivities in the temporal dimension, this section focuses on the variation of national total emissions in each test. Tests influencing both NO$_x$ and CO$_2$ emissions exhibit comparable effects, while prior tests exclusively influence CO$_2$ emissions (Fig. 3). For conciseness, we focus on the $RC_t$ in CO$_2$ emissions in tests here (discussion on NO$_x$ emissions seen in Text S3). The average $RC_t$ of national total emissions are comparable across temporal scales with differences below 1% (lines in Fig. 3, Figs. S8-S9). However, the consistency of $RC_t$ weakens from yearly to monthly to daily scales (increased $1\sigma_t$ as shown by the shadow in Fig. 3). To better characterize the extent of the tests' impact, the discussion here focuses on the $\overline{RC_t}\pm1\sigma_t$ on a daily scale, reflecting the magnitude and consistency of the impact concurrently.

At the national total scale, prior tests (ef_[-10%, -1%], thre_40%/60%, and 4 sectors) influence CO$_2$ emissions consistently over time while leaving NO$_x$ emissions unaffected (Fig. 3). This occurs because these tests only impact sectoral attribution and CO$_2$-to-NO$_x$ emission ratios. Total NO$_x$ emissions are determined in the top-down process before sectoral attribution, thus remaining unchanged (Fig. S1). However, sector-

specific $CO_2$ emissions, derived from $NO_x$ emissions, are influenced due to the varying $CO_2$-to-$NO_x$ emission
ratios among sectors (Fig. S10). A reduction in $NO_x$ EFs increases $rNO_x$, thereby increasing the sectoral $CO_2$-
to-$NO_x$ emission ratios since $CO_2$ EFs are assumed to be unchanged (Eq. 5). This results in a linear elevation
of $CO_2$ emissions in tandem with the decreased $NO_x$ EFs (Fig. S7), with $CO_2$ emission variations reaching
up to 10.7%±0.7% under ef_-10%. Similarly, modifications in threshold values and sector classification alter
the identification of dominant sectors per grid, changing the sectoral attribution. Thre_40%/60% and
4_sectors bring about $\overline{RC_t} \pm 1\sigma_t$ of 0.6%±1.5%, -0.2%±1.7%, and 0.2%±0.8% in $CO_2$ emissions, respectively,
demonstrating their low influence on emission estimates. Despite differences in the magnitude of prior tests'
impacts ( $\overline{RC_t}$ ), they share a consistency at finer temporal scales, with daily $1\sigma_t$ below 4.0%.
Changes in model resolution (Res_2×2.5) introduce the largest variation in estimates among all sensitivity
tests, triggering $\overline{RC_t} \pm 1\sigma_t$ of -1.2%±6.0% in daily $CO_2$ emissions. Its notable inconsistency of impact on the
finer temporal scale ($1\sigma_t > 4.0\%$) can be traced back to its induced spatiotemporally diverse changes in $\beta$
(Figs. S11a and S11b). The overall low estimate of $\beta$ under Res_2×2.5 results in negative $RC_t$, and the uneven
spatial distribution of $\beta$ explains the large $1\sigma_t$.
As for the impact of satellite constraint, the systematic changes such as missing value supplementation
(Trop_fill) or version changes (Trop_v2.3) have a larger impact with daily $CO_2$ emission variations of
1.3%±3.9% and -0.4%±5.9%, while alterations in satellite data quality screening conditions
(Trop_cf/Trop_qa) exert a relatively minor impact on estimates with $\overline{RC_t} \pm 1\sigma_t$ less than 0.5%±1.8%. The
spatiotemporal changes in satellite $NO_2$ retrievals contribute to the inconsistent effects of Trop_fill and
Trop_v2.3 on daily emissions. However, the small $1\sigma_t$ in screening condition tests suggests that the
uncertainty of satellite retrievals has a minor impact on estimates unless there are systematic changes,
possibly because we used the ten-day moving average of satellite observation data to constrain emissions.
Among inversion system parameter tests, the alteration of the reference year (2021_base) exhibits a notable
temporally inconsistent impact, with $\overline{RC_t} \pm 1\sigma_t$ of -0.6%±6.9% in daily $CO_2$ emissions. This inconsistency
can be attributed to the spatiotemporally diverse changes in $\beta$, similar to the model resolution test (Figs. S11c
and S11d). In contrast, changes in $\beta$ ($\beta$_[-20%, 20%]) exert a more notable but consistent impact on estimates,
linearly strengthening as the tested amplitude increases (Fig. S7), with $\beta$_20% triggering variations of
2.6%±3.0% in $CO_2$ emissions. The spatiotemporally uniform changes in $\beta$ act linearly on the inversion
estimate of $NO_x$ emissions (Eq. 1), and then on $CO_2$ emissions. Therefore, their impact remains consistent on
a daily scale.

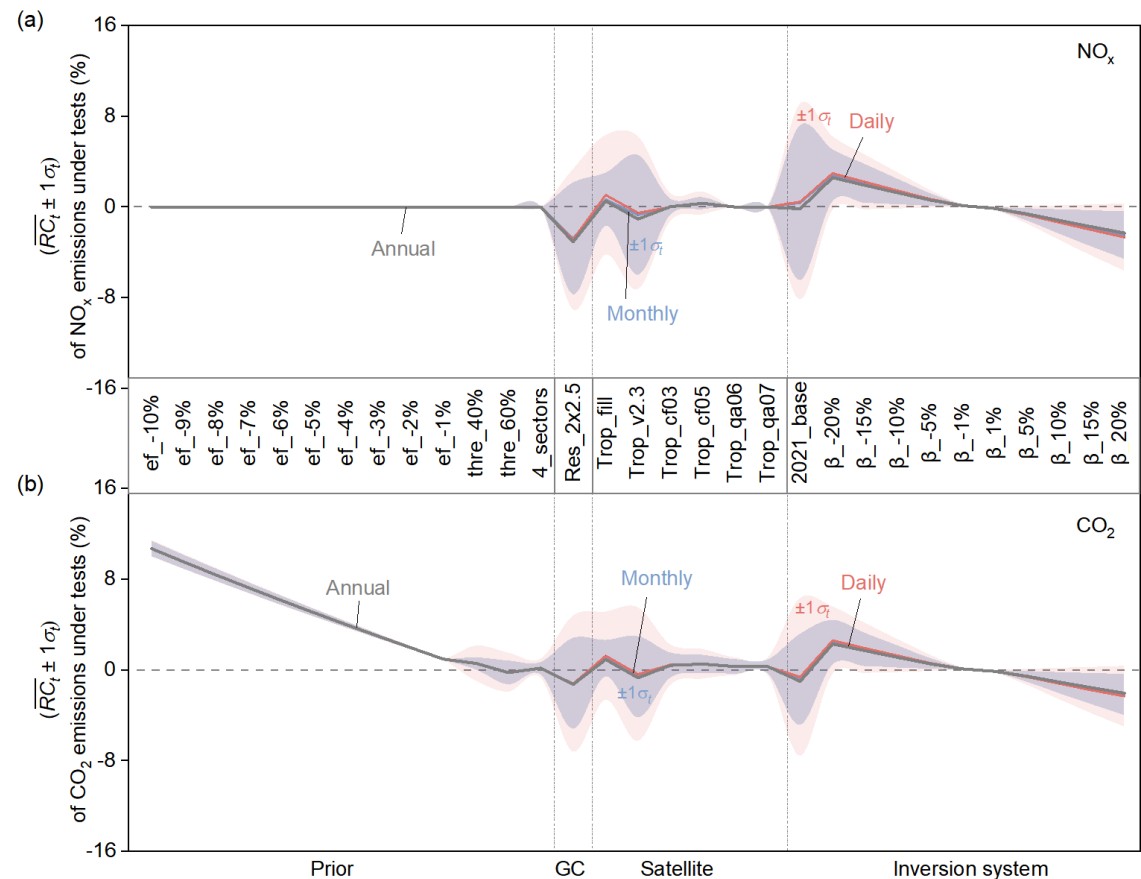


**Figure 3**. **Comparison of the impacts of various tests on national total (a) $NO_x$ and (b) $CO_2$ emissions at different time scales.** Gray lines correspond to the $RC_t$ in annual emissions. Blue lines depict the average $RC_t$ in monthly emissions, with the blue shadow indicating monthly scale $1\sigma_t$. Red lines illustrate the average $RC_t$ in daily emissions, accompanied by the red shadow indicating daily scale $1\sigma_t$.

### 3.3 Emission sensitivity across source sectors

Regarding daily national sectoral $NO_x$ and $CO_2$ emissions, their responses to different sensitivity tests, in terms of both emission magnitude and consistency ($\overline{RC_s} \pm 1\sigma_s$), are largely similar, except for $NO_x$ EFs tests (ef_[-10%, -1%]) (Fig. 4). Therefore, we primarily discuss the impacts of tests on sectoral emissions using $CO_2$ as a representative (refer to Text S4 for discussion on sectoral $NO_x$ emission), and then delve into elucidating the divergent impact of $NO_x$ EFs on sectoral $NO_x$ and $CO_2$ emissions.

Irrespective of $NO_x$ emission factor changes (ef_[-10%, -1%]), industrial and power emissions exhibit greater robustness than transport and residential emissions, which are more susceptible to different configurations. Specifically, residential emissions demonstrate the highest susceptibility to reference year, showing $\overline{RC_s} \pm 1\sigma_s$ of up to -6.7%±7.3% in $CO_2$ emissions in 2021_base test, and exclusively display notable sensitivity to prior tests (4_sectors and thre_40%/60%) compared to other sectors (Fig. 4). In contrast, transport emissions are notably influenced by model resolution, with Res_2×2.5 incurring $CO_2$ emission variations of -7.8%±12.2%. Among all sensitivity tests, the model resolution stands out as the most influential

factor on sectoral emissions, because the resolution of grid cells affects the determination of the dominant source sector.

The overall largest sensitivity of residential emissions to sensitivity tests is potentially attributed to its low proportion to total emissions (Fig. S12). Take thre_40%/60% as an example, lowering the threshold from 50% to 40% results in identifying more grids as residential source dominant. This, in turn, leads to an increase in residential emission proportions when allocating the total TROPOMI-constrained $NO_x$ emissions into sectors and subsequently $CO_2$ emissions. Conversely, fewer grids are assigned as residential-dominant when the threshold rises from 50% to 60%, resulting in lower residential emissions (Fig. S13). The next sensitive sector is transport, particularly vulnerable to model resolution, which may be associated with its characteristics in spatial distribution. Transport-dominant grids, particularly those with truck emissions, are typically located close to industry-dominant grids whose $NO_x$ emissions outweigh those from the transport (Zheng et al., 2020). The use of a coarser horizontal resolution could result in a diminished attribution of emissions to transport (Fig. S14).

The reduction in $NO_x$ EFs (ef_[-10%, -1%]) is the only test impacting sectoral $NO_x$ and $CO_2$ emissions differently. For $NO_x$ emissions, the residential sector shows the strongest sensitivity with $\overline{RC_s} \pm 1\sigma_s$ of up to -9.1%±4.5% under ef_-10%. However, its influence on $CO_2$ emissions is most pronounced in all sectors except residential, with variations of 12.4%±1.1% in $CO_2$ emissions from industry, 11.9%±1.9% from transport, 10.8%±1.2% from power, but only 1.0%±4.9% from residential sectors under ef_-10%. The reduction in $NO_x$ EFs shifts the dominant sector attribution, substantially lowering $NO_x$ emissions from the residential sector due to its vulnerability to these changes, similar to the impact seen with the thre_60%. The other sectoral (industry, transport, and power) $CO_2$ emissions present stronger sensitivity to $NO_x$ EFs tests, linearly correlated with the extent of EFs changes. The decline in sectoral $NO_x$ EFs linearly reduces $rNO_x$ (Eq. 5), raising the corresponding $CO_2$ emissions by increasing sectoral $CO_2$-to-$NO_x$ emission ratios.

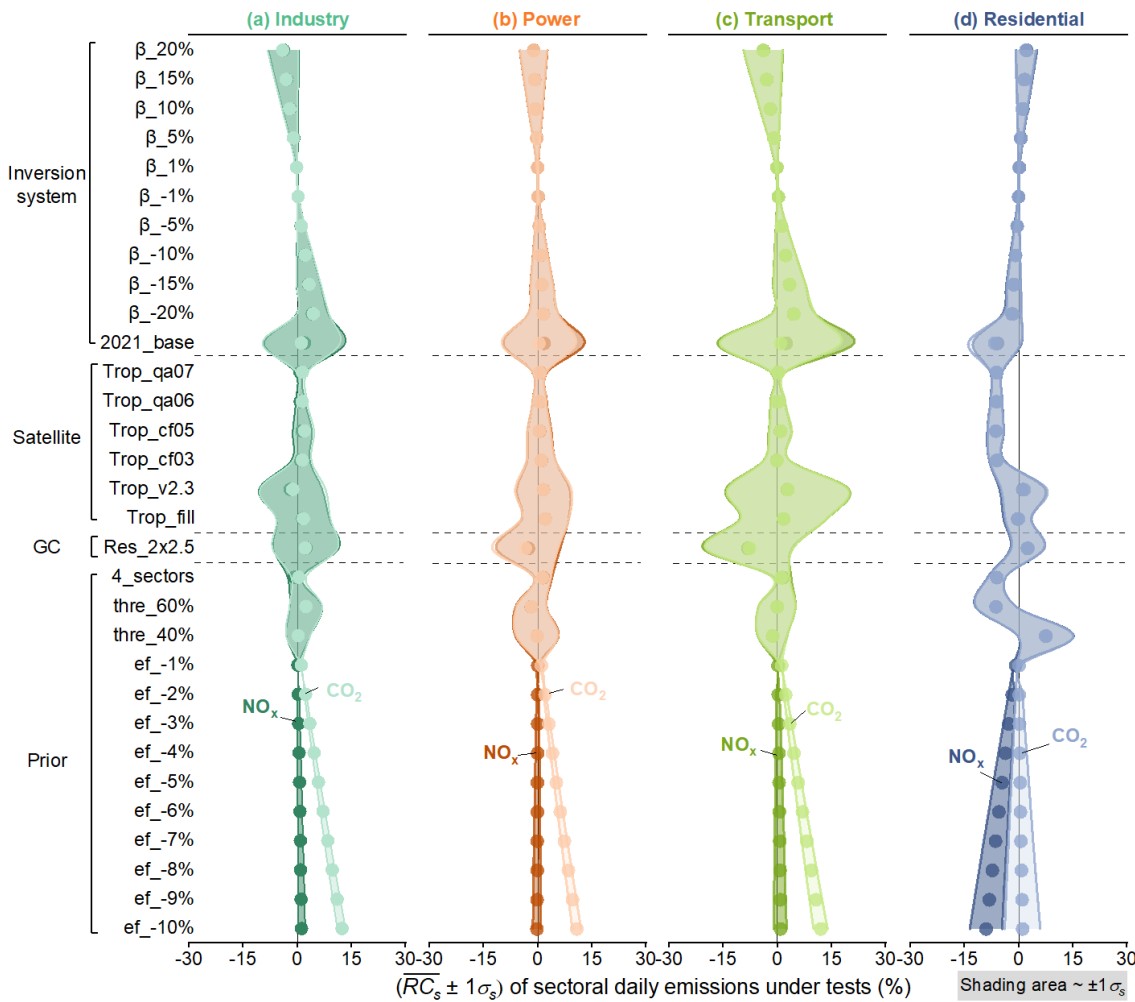

**355**

**Figure 4**. **Response of sectoral national NOₓ and CO₂ emissions to different sensivity tests on a daily scale.** From left to right, the panels correspond to the (**a**) industry, (**b**) power, (**c**) transport, and (**d**) residential source sectors, as the label notes. The dots inside each figure are the average $RC_s$ of daily NOₓ (deep color) and CO₂ (light color) emissions incurred by corresponding tests. The shading area indicates the $1\sigma_s$ of $RC_s$ of daily sectoral emissions in different tests.

### 3.4 Emission sensitivity at subnational scales

Refining spatial coverage from national to subnational level (i.e., province) reveals that factors causing inconsistent impacts over finer time scales also tend to induce inconsistent impacts on more granular spatial regions (Fig. 5). On the annual total scales, the $RC_p$ of NOₓ and CO₂ emissions at the provincial scale closely resemble each other under most sensitivity tests, except for prior tests that only influence CO₂ emissions (Fig. S15). When comparing across provinces, the sensitivity of emissions to tests correlates with the size of the provincial area, with smaller regions exhibiting greater susceptibility. Shanghai, the smallest provincial-level administrative unit in China in terms of area, experiences the largest $RC_p$ throughout China in nearly all tests. Conversely, Inner Mongolia, one of China's top three largest provinces, undergoes the minimum $RC_p$ in all tests. Under Res_2×2.5, the $RC_p$ of annual total NOₓ and CO₂ emissions in Shanghai are 19.6% and 22.6%, respectively, while in Inner Mongolia, they are -3.2% and -3.3%. Employing a resolution of 2°×2.5° in

Shanghai is impractical in real-world applications, as it would result in fewer than two grids covering the area. Henan also encounters substantial $RC_p$ under Res_2×2.5, reaching as high as -15.8% and -12.4% in annual total $NO_x$ and $CO_2$ emissions. This could be attributed to its proximity to Shandong, a province with approximately twice the emissions of Henan, making Henan particularly sensitive to the changes in model resolution due to the overlapping grid cells. It is noteworthy that Guizhou exhibits the highest sensitivity to satellite constraint, with $RC_p$ reaching up to 11.9% and 11.8% in annual total $NO_x$ and $CO_2$ emissions under Trop_v2.3. This sensitivity is attributed to the high cloudiness of the Yunnan-Guizhou Plateau, causing satellite observations to be highly uncertain over Guizhou (Wang et al., 2023; Li et al., 2021; Cai et al., 2022).

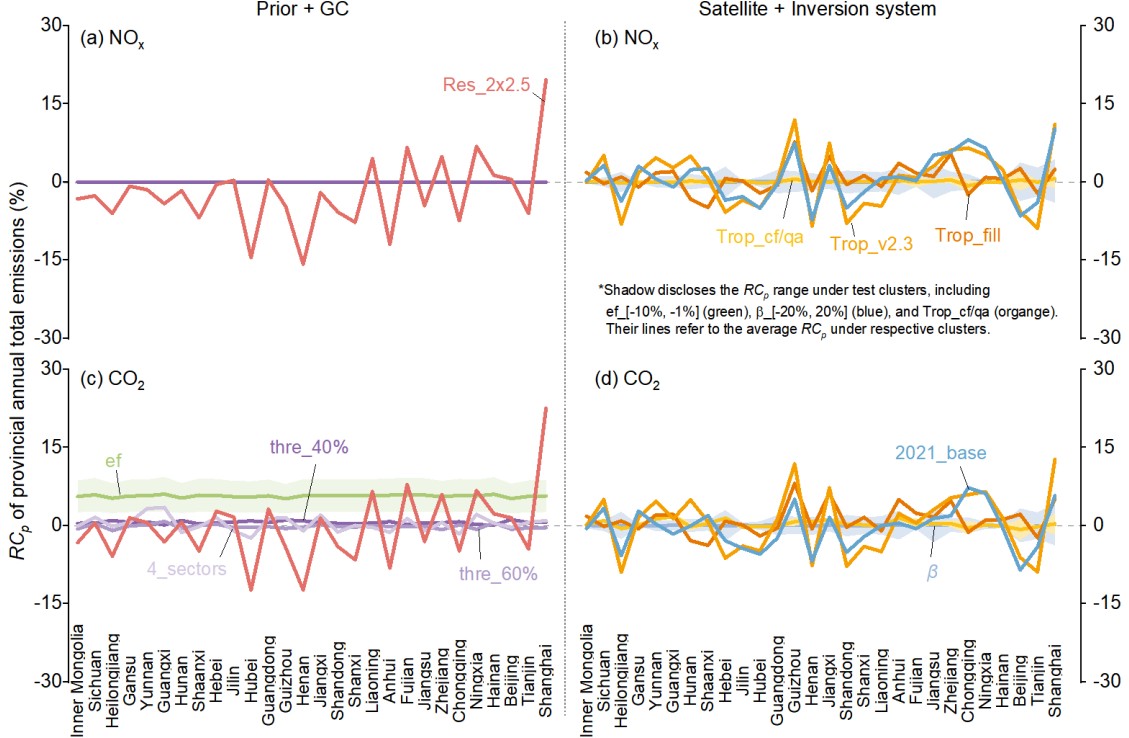

**Figure 5**. **Response of provincial annual total $NO_x$ and $CO_2$ emissions to different tests.** (**a**) and (**b**) show $RC_p$ of $NO_x$ emissions incurred by tests. (**c**) and (**d**) are plotted for $CO_2$ emission as (**a**) and (**b**). Lines refer to the $RC_p$ caused by the corresponding test or the averaged $RC_p$ caused by corresponding test clusters (ef_[-10%, -1%] and β_[-20, 20%]), and the shadow refers to the $RC_p$ range in test clusters. Only provinces with enough TROPOMI observations are shown here (i.e., grids with $NO_2$ TVCDs larger than $1×10^{15}$ molecules/cm$^2$ cover more than 90% of anthropogenic $NO_x$ emissions within provinces). The provinces are arranged by area.

To further investigate the daily total emission response ($\overline{RC_r} \pm 1\sigma_r$) to tests at the regional scale, we select and analyze Jing-Jin-Ji clusters (JJJ, including Beijing, Tianjin, and Hebei), Inner Mongolia, Yangtze River Delta clusters (YRD, including Shanghai, Zhejiang, and Jiangsu), and Guangdong (the location of the Pearl River Delta). These regions respectively represent an industrialized region with high population density, an industrialized region with sparse population density, and two major economic development zones with high population density in China (Fig. 6). Geographically, these regions span North China (JJJ and Inner Mongolia), East China (YRD), and South China (Guangdong), thereby covering different meteorological and

geographic factors. Overall, the $\overline{RC_r} \pm 1\sigma_r$ of daily regional emissions are similar for $NO_x$ and $CO_2$ except for
ef_[-10%, -1%], resembling their daily national emission responses (Fig. 3). The $\overline{RC_r} \pm 1\sigma_r$ of daily regional
emissions is especially notable in YRD and Guangdong (southern part of China). This could be attributed to
the relatively low $NO_2$ concentration in southern China (Fig. S4), making them particularly sensitive to spatial
variations in parameters, such as the $\beta$ in 2021_base (Fig. S11) and $NO_2$ TVCDs in Trop_v2.3 test. Besides,
the cloud fraction is higher in southern China, introducing larger uncertainties in remote sensing (Liu et al.,
2019; Latsch et al., 2022). The emission responses to prior and β_[-20%, 20%] tests are close for these four
regions, particularly in the prior tests, suggesting that these impacts on emissions are less dependent on
geographic factors.

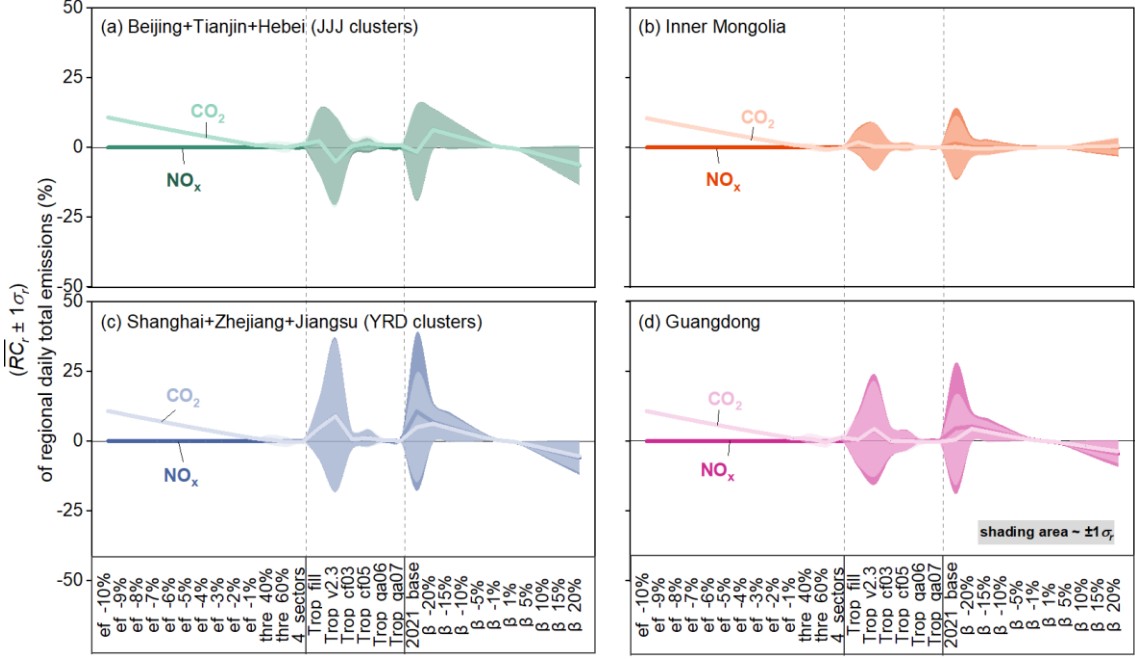


**Figure 6**. **Response of regional total $NO_x$ and $CO_2$ emissions to tests on a daily scale.** (**a**), (**b**), (**c**), and (**d**)
show the $\overline{RC_r} \pm 1\sigma_r$ of daily $NO_x$ (deep color) and $CO_2$ (light color) emissions in different tests in Jing-Jin-Ji
clusters (Beijing, Tianjin, and Hebei), Inner Mongolia, Yangtze River Delta clusters (Shanghai, Zhejiang,
and Jiangsu), and Guangdong. The shading area inside each figure refers to the corresponding $1\sigma_r$. It is worth
noting that the Res_2×2.5 test is not shown here since the resolution of 2°×2.5° proves too coarse for certain
regions, rendering it unrealistic for real-world applications. The result containing Res_2×2.5 is present in SI
as Fig. S16 for reference.
**4 Discussion**
This study delineates an approximate spectrum of uncertainties inherent in deriving conclusions of varying
precision with our air pollution satellite sensor-based $CO_2$ emission inversion system. When interpreting
conclusions based on the emission data derived from such an inversion system, it is practical and imperative
to aggregate emissions across different dimensions to fulfill specific usage requirements. Direct utilization
of data with all fine-grained resolutions at temporal, sectoral, and spatial dimensions poses challenges. If

adhering to a variation tolerance of 5%, the reliability of annual national $NO_x$ and $CO_2$ emissions is established in most cases. Notably, careful attention is needed when selecting model resolution and attributing sectoral emissions. Expanding the tolerance to 10%, which is still below the conventional bottom-up method's uncertainty range of 13%-37% (Zhao et al., 2011; Huo et al., 2022), renders annual regional or daily national emissions robust from an average perspective. Nevertheless, meticulous scrutiny is advised when drawing conclusions based on daily sectoral or daily regional emissions, especially in specific regions (e.g., Shanghai, Guizhou). The large uncertainty of daily sectoral emission is typically observed in other emission datasets, such as Carbon Monitor (up to 40% uncertainty) (Liu et al., 2020c; Huo et al., 2022). Further liberalizing the tolerance to 25%, which is quite uncertain for scientific and policy-making purposes, the majority of conclusions derived from our estimates stand as reliable. The extensive tolerance range primarily stems from regional emissions, posing a challenging issue for many emission inversion techniques. For example, the uncertainty in $NO_x$ emissions derived from the 2D MISATEAM (chemical transport Model-Independent SATellite-derived Emission estimation Algorithm for Mixed-sources) method is approximately 20% for large and mid-size US cities (Liu et al., 2023), and the uncertainty for daily $NO_x$ and $CO_2$ emissions based on the superposition model ranges from 37% to 48% on a city scale (Zhang et al., 2023). Notably, remarkable advancements have been achieved in estimating subnational $CO_2$ emissions through $CO_2$-observing satellites, such as sectoral $CO_2$ assessments with OCO-3 (Roten et al., 2023), and urban emission optimizations utilizing the Orbiting Carbon Observatory-2 (OCO-2) (Yang et al., 2020; Ye et al., 2020). Yet, reducing uncertainties at subnational scales remains an ongoing challenge.

This study paves the way for the continuous improvement of the current air pollution satellite sensor-based $CO_2$ emission inversion system. Firstly, prioritizing a nimble and appropriate horizontal resolution is crucial for establishing accurate localized relationships between $NO_2$ TVCDs and $NO_x$ emissions, contributing to improved $NO_x$ and $CO_2$ emission estimations from temporal, sectoral, and spatial perspectives. Secondly, the more accurate satellite observation is conducive to reducing the uncertainty in final results, presenting increasing promise with advancements in remote sensing technology. Besides, the progress in multi-species synchronous observations through satellite and aircraft platforms offers alternative verification for multi-species emission inversion, such as the Copernicus Anthropogenic Carbon Dioxide Monitoring constellation (CO2M) (Sierk et al., 2021). Thirdly, the reliability of sectoral $NO_x$ EFs changes, which determine $CO_2$-to-$NO_x$ emission ratios, is essential for the accurate conversion from $NO_x$ to $CO_2$ emissions. This underscores the need to acquire more accurate $NO_x$ EFs. While obtaining on-site measurements of $CO_2$-to-$NO_x$ emission ratios is challenging, efforts are underway to enhance its configuration. An iterative modification of $NO_x$ EFs within the current system could be incorporated, minimizing the gap between bottom-up updated and TROPOMI-constrained sectoral $NO_x$ emissions to below 2%. This approach yields more accurate $CO_2$-to-$NO_x$ emission ratios and $CO_2$ emissions (Fig. S17). The optimized $CO_2$ emission change from 2021 to 2022 is +0.6%, reflecting a more precise representation of the growth in fossil fuel consumption (+1.9%). Fourthly, utilizing a more refined approach to determine dominant sectors at a grid level can reduce the uncertainty of small-contributing sectoral emissions, particularly in the residential sector. These enhancements will improve

the system's accuracy in estimating emissions across all dimensions, positioning it as a valuable tool for simultaneous inversion-based monitoring of greenhouse gas and air pollutants emissions, ultimately supporting a strategic roadmap for the vision of clean air and climate warming mitigation.

*Code and data availability.* The source code of the GEOS-Chem model is available at https://geoschem.github.io/. The prior $NO_x$ and $CO_2$ emissions of 2019 MEIC (v1.4) are available at http://meicmodel.org.cn/?page_id=541&lang=en. The v2.4.0 TROPOMI $NO_2$ column concentrations are publicly available at https://www.temis.nl/airpollution/no2col/no2regio_tropomi.php. The activity level data of China from 2019 to 2022 including the industrial production of cement, iron, thermal electricity, etc., are available at https://data.stats.gov.cn/english/easyquery.htm?cn=C01.

*Supplement.* The supplement related to this article is available online.

*Author Contributions.* Bo Zheng designed the research and led the analysis. Hui Li performed the simulation, analyzed the data, and created the graphs. Bo Zheng, Jiaxin Qiu, and Hui Li wrote the manuscript.

*Competing interests.* The authors declare that they have no conflict of interest.

*Acknowledgements.* The authors thank the editor and the anonymous referees for helpful comments that have improved the paper.

*Financial support.* This work was supported by the National Key R&D Program of China (2023YFC3705601) and National Natural Science Foundation of China (Grant No. 42375096).

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
