# Peer review of "Air pollution satellite-based CO2 emission inversion: system"

_EGUsphere, 2024_

## Author Comment (AC1)

**Reviewer: 1**

This study presents a sensitivity analysis for a new inversion technique that estimates $CO_2$ emissions from co-emitted air pollutants ($NO_2$). The inversion methodology is an interesting way of bypassing challenges in $CO_2$ remote sensing and takes advantage of the relative ease of $NO_2$ detection with remote sensing relative to $CO_2$. While the methodology has been presented elsewhere in the literature with useful applications in real-time greenhouse gas monitoring, a rigorous assessment of its sensitivity to the different input variables is valuable for optimisation moving forwards. The separation of sensitivities into spatial, temporal etc. is particularly nice, especially as we strive for greater and greater resolution in these dimensions. This makes it easy to understand the limitations for specific use cases. In general, the manuscript is of high written and visual quality, and the analysis is sound. I have a few minor comments surrounding the prior $NO_x$ emissions as well as some suggestions below.

**Response:**

We express our gratitude to the referee for providing constructive and positive feedback on our manuscript. Below, we offer detailed responses addressing each point raised

1. Line 89: What are the sector specific scaling factors? Which sectors and by how much they are scaled (inaccurate) is one of the most valuable outputs of this kind of methodology from a $NO_x$ standpoint. It would be nice to see a plot displaying this in the SI.

**Response:**

We have added Fig. S2 in SI displaying sectoral correction factors, which mainly range from 0.5 to 1.5, and a brief explanation of this in Lines 127-128.

Lines 127-128: "The overall sectoral correction factors mainly range from 0.5 to 1.5 (Fig. S3)."

[Figure]

**Figure S3. The comparison between bottom-up and TROPOMI-constrained sectoral $NO_x$ emissions (Base inversion).** The upper panel shows the sectoral

30    correction factors.

2. Line 94: I have concerns about the accuracy of $CO_2$-$NO_x$ emission ratios. My
knowledge of Chinese emissions inventories is poor. However, in European emissions
inventories emission factors for $NO_x$ can be very outdated. Perhaps this is taken into
35    account with the scaling factors discussed in Line 89. I think a discussion of the
emissions inventory in addition to the sector specific scaling factors, and even a
comparison with other international emissions inventories would be useful e.g.
EEA/EMEP, US EPA.

**Response:**

40    The $CO_2$-to-$NO_x$ emission ratios (ERs) from the 2019 MEIC inventory are considered
relatively reliable, having been validated in previous simulations (Zheng et al., 2021;
Zheng et al., 2020). Although the changes in these ratios since 2019 remain uncertain,
we assumed a reduction in $NO_x$ emission factors (EFs) while keeping $CO_2$ EFs constant
from 2019 to 2022 to estimate updated $CO_2$-to-$NO_x$ ERs for 2022. This assumption
45    aligns with the ongoing emission control measures implemented by the Chinese
government. To assess the influence of this assumption, we performed sensitivity tests
on varying $NO_x$ EF reduction levels, which demonstrated a significant impact on $CO_2$
emissions. Additionally, a comparison of our $CO_2$-to-$NO_x$ ERs with other international
inventories (EDGAR and CEDS) shows our values fall within the mid-range.

50    To make these information clearer, we have added some explanation about the $CO_2$-to-
$NO_x$ emission ratios (ERs) in Lines 135-139 in the Manuscript and a detailed discussion
of $CO_2$-to-$NO_x$ ERs in Text S2 in SI, which includes the method of ERs updates, the
sensitivity tests on this settings, and comparison with international emission inventories
in China (EDGAR and CEDS).

55    Lines 135-139: "The $CO_2$-to-$NO_x$ emission ratios in 2022 are updated by reducing $NO_x$
emission factors (EFs) while keeping $CO_2$ EFs unchanged based on 2019 MEIC. The
default assumption that the reduction rate halves annually is due to the limited potential
for further reductions. In contrast, the $CO_2$ EFs are assumed to remain unchanged, as
they are primarily determined by fuel type and combustion conditions (Cheng et al.,
60    2021) (details seen in Text S2)."

**Text S2. $CO_2$-to-$NO_x$ emission ratios**

In this inversion system, the $CO_2$-to-$NO_x$ emission ratios (ERs) are initially derived
from the 2019 MEIC inventory, then updated for the target year (2022 in this study) by
assuming a specific reduction in $NO_x$ EFs by sector while keeping $CO_2$ EFs constant.
65    This approach aligns with the ongoing decline in $NO_x$ emissions due to pollution control
measures, while $CO_2$ emissions remain more closely tied to fuel type and combustion
conditions (Text S1). Accordingly, the $CO_2$-to-$NO_x$ ERs are dependent on the reduction
ratio of $NO_x$ EFs in this system (represented by the $rNO_{x\,s,i,y}$ in Eq. 5).

The reduction ratio of $NO_x$ EFs first influences the disaggregation of total $NO_x$
70    emissions to sectors, and then affects the sector-specific conversion from $NO_x$ to $CO_2$
emissions. To evaluate this impact, we set a gradient test with a $NO_x$ EFs reduction
range from 1% to 10% (ef_[-10%, -1%]). Results indicate a notable impact on $CO_2$
emissions, affecting annual national $CO_2$ totals by up to 10.7% (Details discussed in
Manuscript). This finding emphasizes the need for a more precise approach to setting
75    $NO_x$ emission reduction ratios in future refinements, such as incorporating an iterative

adjustment within the bottom-up process to better align bottom-up and TROPOMI-constrained sectoral $NO_x$ emissions (as mentioned in the Discussion).

We further compare the $CO_2$-to-$NO_x$ ERs of MEIC with some international inventories, including the Emissions Database for Global Atmospheric Research (EDGAR, https://edgar.jrc.ec.europa.eu/dataset_ap81) (Crippa et al., 2020) and the Community Emissions Data System (CEDS) (Mcduffie et al., 2020), for the year 2019. Given the different categorization structures in these inventories, we focus on comparing the overall $CO_2$-to-$NO_x$ ERs, which are 493.7 for MEIC, 571.5 for EDGAR, and 462.6 for CEDS. The emission factors in MEIC are more spatially and sectorally refined for China, making its $CO_2$-to-$NO_x$ ERs more representative of China-specific emissions (Zheng et al., 2018).

3. Line 104: Where does this 40% reduction come from? This is not discussed in the text.

**Response:**

The 40% reduction in simulation is used to quantify the response of $NO_2$ concentration to the changes in anthropogenic $NO_x$ emission ($\beta$), building on previous works. In our previous tests, this perturbation magnitude seems to have a limited impact on final estimates within the tested range of 30-50%. We have added a brief explanation in Lines 110-112 in Manuscript.

Lines 110-112: "The 40% reduction was selected after a series of sensitivity tests, which demonstrated that this perturbation level exerts a limited impact on the $\beta$ estimates (Zheng et al., 2020)."

4. Line 135: How do the sector scaling factors in Line 89 compare to the -1 to -10 % gradient system? Is -10 % a high enough threshold? Why do you only consider a negative range?

**Response:**

China has enforced stringent emission controls on anthropogenic $NO_x$ emissions for decades, achieving substantial reductions. Since 2012, $NO_x$ emissions in China have been consistently decreasing; however, as reduction potential diminishes, the rate of decrease has recently begun to slow (Li et al., 2023). For instance, between 2013 and 2017, the annual reduction rate in $NO_x$ emissions was around 5.2%, but it slowed to 3.2% between 2018 and 2020 (Geng et al., 2024). Consequently, a 10% reduction in $NO_x$ emission factors now represents a challenging and idealized scenario.

Regarding the exclusive consideration of negative trends, ongoing emission control policies and actions further underscore the continuous downward trajectory of $NO_x$ emissions, as consistently reported by recent studies (Geng et al., 2024; Li et al., 2023). Thus, a downward shift in $NO_x$ emission factors over time is more consistent with the current policies.

**Grammatical:**

5. Line 11: Suggest removal of "to prevent irreversible damage". Not needed and air pollution is generally not irreversible.

**Response:**

120  We have removed the "to prevent irreversible damage" in Line 12 (original 11) as suggested.

6. Line 24: add "the" after "example,".

**Response:**

125  We have added "the" in Line 24 as suggested.

7. Line 28: Suggest change to "how much, where, and by what activity pollutants are released…".

**Response:**

130  We have modified the Line 29 (original 28) as suggested, as shown below:

Line 29: "The knowledge of emissions, i.e., how much, where, and by what activity pollutants are released into the atmosphere,"

8. Line 61: Suggest change "Our analytical endeavour" to "This study investigates".

135  **Response:**

We have changed the to "This study investigates" in Line 71 (original 61) as suggested.

Line 71: "This study investigates how emission outcomes respond to a variety of sensitivity assessments across temporal, sectoral, and spatial dimensions."

140  9. Line 217: Suggest removal of "(all columns except the first one)". No need to clarify.

**Response:**

We have removed "all columns except the first one" in Line 251 (original 217) as suggested.

145  10. Line 258: Suggest replacement of "least" with "low".

**Response:**

We have replaced the "least" with "low" in Line 291 (original 258) as suggested.

**Figures/Tables:**

150  11. Fig S5: misspelling of national in y-axis label.

**Response:**

We have corrected the spelling of "national" in the y-axis label in Fig. S7 (original Fig.

[Figure]

**Figure S7. Sensitivity of annual national total NOₓ and CO₂ emissions to β and NOₓ emission factor.** (a) and (c) present the estimated NOₓ emissions under a ten-level gradient for β and emission factor variations. (b) and (d) are plotted for $CO_2$ emissions as (a) and (c).

12. Fig S11: It would be good to see this plot vs temperature. Why is there such a big drop in March? If it is correlated well, this would be a good verification of the system.

**Response:**

We have added the heating degree day (HDD) in Fig. S13 (original S11), which shows a good agreement with the residential emission dynamics.

[Figure]

**Figure S13. The comparison of proportion attributing total TROPOMI-constrained NO$_x$ emissions to the residential sector.** Black, red, and blue lines refer to the Base, thre_40%, and thre_60% inversions, respectively. The upper panel displays the temporal variation of the national average heating degree day.

13. Table 1: Please can you clarify what you mean by "reduction ratio of NO$_x$ EFs halves annually"?

**Response:**

The "reduction ratio of NO$_x$ EFs halves annually" means that each year's reduction rate for NO$_x$ EFs is set to decrease by half compared to the previous year. For example, if the reduction of NO$_x$ EFs from 2019 to 2020 was 4%, the reduction from 2020 to 2021 would be set at 2%.

We have added an explanation in the Note below Table 1 in Lines 149-150:

Lines 149-150: "*Each year's reduction rate for NO$_x$ EFs is set to decrease by half compared to the previous year. For example, if the reduction of NO$_x$ EFs from 2019 to 2020 was 4%, the reduction from 2020 to 2021 would be set at 2%."

**Reference:**

Cheng, J., Tong, D., Liu, Y., Bo, Y., Zheng, B., Geng, G., He, K., and Zhang, Q.: Air quality and health benefits of China's current and upcoming clean air policies, Faraday Discussions, 226, 584-606, https://doi.org/10.1039/D0FD00090F, 2021.

Crippa, M., Solazzo, E., Huang, G., Guizzardi, D., Koffi, E., Muntean, M., Schieberle, C., Friedrich, R., and Janssens-Maenhout, G.: High resolution temporal profiles in the Emissions Database for Global Atmospheric Research, Scientific Data, 7, 121, 10.1038/s41597-020-0462-2, 2020.

Geng, G., Liu, Y., Liu, Y., Liu, S., Cheng, J., Yan, L., Wu, N., Hu, H., Tong, D., Zheng, B., Yin, Z., He, K., and Zhang, Q.: Efficacy of China's clean air actions to tackle $PM_{2.5}$ pollution between 2013 and 2020, Nature Geoscience, 17, 987-994, 10.1038/s41561-024-01540-z, 2024.

Li, S., Wang, S., Wu, Q., Zhang, Y., Ouyang, D., Zheng, H., Han, L., Qiu, X., Wen, Y., Liu, M., Jiang, Y., Yin, D., Liu, K., Zhao, B., Zhang, S., Wu, Y., and Hao, J.: Emission trends of air pollutants and $CO_2$ in China from 2005 to 2021, Earth Syst. Sci. Data, 15, 2279-2294, https://doi.org/10.5194/essd-15-2279-2023, 2023.

McDuffie, E. E., Smith, S. J., O'Rourke, P., Tibrewal, K., Venkataraman, C., Marais, E. A., Zheng, B., Crippa, M., Brauer, M., and Martin, R. V.: A global anthropogenic emission inventory of atmospheric pollutants from sector- and fuel-specific sources (1970–2017): an application of the Community Emissions Data System (CEDS), Earth Syst. Sci. Data, 12, 3413-3442, 10.5194/essd-12-3413-2020, 2020.

Zheng, B., Zhang, Q., Geng, G., Chen, C., Shi, Q., Cui, M., Lei, Y., and He, K.: Changes in China's anthropogenic emissions and air quality during the COVID-19 pandemic in 2020, Earth Syst. Sci. Data, 13, 2895-2907, https://doi.org/10.5194/essd-13-2895-2021, 2021.

Zheng, B., Geng, G., Ciais, P., Davis, S. J., Martin, R. V., Meng, J., Wu, N., Chevallier, F., Broquet, G., Boersma, F., van der A, R., Lin, J., Guan, D., Lei, Y., He, K., and Zhang, Q.: Satellite-based estimates of decline and rebound in China's $CO_2$ emissions during COVID-19 pandemic, Science Advances, 6, eabd4998, https://doi.org/10.1126/sciadv.abd4998, 2020.

Zheng, B., Tong, D., Li, M., Liu, F., Hong, C., Geng, G., Li, H., Li, X., Peng, L., Qi, J., Yan, L., Zhang, Y., Zhao, H., Zheng, Y., He, K., and Zhang, Q.: Trends in China's anthropogenic emissions since 2010 as the consequence of clean air actions, Atmos. Chem. Phys., 18, 14095-14111, https://doi.org/10.5194/acp-18-14095-2018, 2018.

---

## Author Comment (AC2)

**Reviewer: 2**

This work presents a robust uncertainty analysis for an established mass balance inversion scheme capable of inferring $CO_2$ emissions from TROPOMI's $NO_2$ measurements. While I do not take issue with the results presented in this manuscript, I found myself carefully re-reading the text multiple times to try and find information I felt to be crucial to the methodology. Some of the information was found after multiple readings while some remained elusive. The omission of certain points in the methodology section and its lack of organization made reading difficult. I have listed my comments, both major and minor, below.

**Response:**

We express our gratitude to the referee for constructive remarks regarding our manuscript. Below, we provide detailed responses addressing each point raised.

**Major Comments**

1. In Lines 38-40, the text mentions the "co-emissions characteristics in time and space" of $NO_2$ and $CO_2$ emissions, leveraging the linear relationship between the two (Yang et al., 2023; Fig. 1). However, in other work by the author (Li and Zheng, 2024; Paper highlight #2), they state that $NO_x$ and $CO_2$ are inversely proportional (at least during COVID lockdowns). Upon first reading, this seems like a contradiction. Perhaps the relationship between $NO_x$ and $NO_2$ emissions should also be discussed in the introduction, near lines 38-40. At least conceptually highlight the conversion from TROPOMI $NO_2$ to $NO_x$ here, particularly how works (eqn. 2).

**Response:**

Anthropogenic $NO_x$ and $CO_2$ are co-emitted, yet their sector-specific emission ratios differ, leading to potentially distinct trends in their total emissions. Specifically, emission controls implemented by the Chinese government have reduced $NO_x$ emission factors (EFs) over time, while $CO_2$ EFs have remained stable, primarily due to their dependence on fuel type and combustion conditions. Thus, given the asynchronous changes in activity levels, $NO_x$ EFs, and $CO_2$ EFs, differing trends in overall $NO_x$ and $CO_2$ emissions are possible.

In the $NO_x$ family, NO is the primary species emitted and undergoes rapid conversion to $NO_2$, which is also the component detectable by most satellites. Therefore, $NO_2$ effectively serves as a proxy for $NO_x$ emissions in inversion studies. $NO_x$ is a short-lived species, making its concentrations highly sensitive to emission sources. This enables the use of mass-balance methods to estimate $NO_x$ emissions, which rely on the assumption of a linear relationship between $NO_2$ columns and local $NO_x$ emissions (Cooper et al., 2017; Mun et al., 2023; Martin et al., 2003).

We have added some explanations in Lines 42-46, Lines 48-50, and Lines 98-100 in Manuscript.

Lines 42-46: "$NO_2$ forms rapidly after NO is emitted from sources and is also the primary nitrogen oxide detectable by most satellites (Ye et al., 2016). This makes $NO_2$ a reliable and widely adopted proxy in nitrogen oxides ($NO_x = NO_2+NO$) emission inversions. However, the co-emission of $NO_x$ and $CO_2$ does not imply synchronized

trends in their emissions, as the $CO_2$-to-$NO_x$ emission ratios and activity trends vary
across different sectors (Li and Zheng, 2024)."

Lines 48-50: "This short lifespan of $NO_2$ facilitates mass-balance approaches for
estimating $NO_x$ emissions, which rely on the assumption of a linear relationship
between $NO_2$ columns and local $NO_x$ emissions (Cooper et al., 2017; Mun et al., 2023;
Martin et al., 2003)."

Lines 98-100: "A critical step in this process was establishing a linear relationship
between $NO_2$ tropospheric vertical column densities (TVCDs) and anthropogenic $NO_x$
emissions under the mass balance assumption (Eq. 2) through GEOS-Chem simulation
(v12.3.0, https://geoschem.github.io/) at a horizontal resolution of 0.5°×0.625°."

2. Lines 46–50 claim that space-based observers of $NO_2$ have surpassed $CO_2$ observers
in revisits, spatial resolution, and coverage. However, I question at least some aspects
of this statement. While TROPOMI has a daily revisit time, it is restricted to a ~1:30
pm overpass time. The $CO_2$-observing OCO-3 instrument provides coverage at
different times throughout daytime hours, providing the potential to elucidate diurnal
emissions (albeit with a ~3 day revisit time). Additionally, OCO-3 has a higher spatial
resolution than TROPOMI, on the order of 2km x 2km. Thus, it is my opinion that Lines
46-50 make unfair statements by not acknowledging the benefits of the OCO-3
instrument.

**Response:**

We have rephrased this sentence acknowledging the development of $CO_2$ satellites in
Lines 53-60.

Lines 53-60: "Moreover, remote sensing technologies for $NO_2$ remain generally more
mature, as indicated by the broader coverage and improved signal-to-noise ratio in
column concentration observation (Macdonald et al., 2023; Cooper et al., 2022). Recent
advancements in $CO_2$ satellite technology are promising, such as the Orbiting Carbon
Observatory-3 (OCO-3), which can generate $CO_2$ maps with a resolution of up to 1.6
km × 2.2 km and monitor $CO_2$ columns at different times throughout the daytime to
elucidate diurnal emission patterns (Taylor et al., 2023), while its spatial coverage may
not be sufficient for large-area inversions at high temporal resolution."

3. Furthermore, this paper does not take into account the most recent efforts to measure
sector-specific $CO_2$ emissions at a sub-annual scale (see Roten et al., 2023 for example).
The title of this work "Air Pollution Satellite-based $CO_2$ Emission Inversion: System
Evaluation, Sensitivity Analysis, and Future Perspective" suggests that the focus will
be on the uncertainty/error of the posterior $CO_2$ estimates. There is little discussion of
the current uncertainties of these measurements, approximated with "direct" $CO_2$
observations, not $NO_x$. Results should be presented in light of recent OCO-2, OCO-3,
etc work. Several publications include city- and sub-city-level emission estimates using
$CO_2$ observations, not $CO_2$ approximations. Consider uncertainties determined by Yang
et al., 2020 and Ye et al., 2020 presenting constraints on $CO_2$ emissions using $CO_2$
observations directly. (Of course, results presented here are sector-specific. Yang and
Ye are not.)

(Roten: https://agupubs.onlinelibrary.wiley.com/doi/full/10.1029/2023GL104376)

(Yang: https://agupubs.onlinelibrary.wiley.com/doi/full/10.1029/2019JD031922)

(Ye: https://agupubs.onlinelibrary.wiley.com/doi/full/10.1029/2019JD030528)

**Response:**

We have added some discussion of $CO_2$-observing $CO_2$ emission inversion in the Discussion Section (Lines 432-436).

Lines 432-436: "Notably, remarkable advancements have been achieved in estimating subnational $CO_2$ emissions through $CO_2$-observing satellites, such as sectoral $CO_2$ assessments with OCO-3 (Roten et al., 2023), and urban emission optimizations utilizing the Orbiting Carbon Observatory-2 (OCO-2) (Yang et al., 2020; Ye et al., 2020). Yet, reducing uncertainties at subnational scales remains an ongoing challenge."

4. The authors should consider reordering the methodology sections. For example, moving 2.1 (Base Inversion) after 2.2.4 and updating the text would let Sections 2.2.1-2.2.4 provide more context in the presentation of equations 1-4. The way the methodology is currently presented is quite confusing. I found myself rereading these sections multiple times to really understand what was going on. Several of these sections are missing helpful information. For example, the section titled "Prior Emission Inventory" (2.2.1) never actually mentions the name of the inventory being used. This made tracking down information difficult throughout my reading of the manuscript. Furthermore, for readers who are unfamiliar with the MEIC inventory, a figure like Fig. 1 of Roten et al., 2023 would be helpful.

**Response:**

The original structure of the Methods section is organized as follows: we begin with an overview of the inversion methodology, using the Base inversion as a foundational example. This is followed by a detailed explanation of the rationale and methodology behind the sensitivity tests. To enhance clarity in discussing the total of 31 tests, we categorized the tested parameters into four classes based on their functions within the system. These categories include changes in prior updates, coarser model resolution, modifications to satellite observational constraints, and other systematic parameters, as depicted in Figure 1. To clarify our approach and reduce misleading, we have added more details about the methodology and re-order them in Section 2.1 (please refer to the Manuscript to track the changes in Section 2.1), added some explanatory notes, and revised the subtitles of Sections 2.1 and 2.2 as follows:

Sub-titles of 2.1: "2.1 Inversion methodology and Base inversion"

Line 87: "We use the Base inversion as a case to provide a detailed explanation of this inversion system."

Sub-titles of 2.2: "Sensitivity settings"

Line 152-158 "The sensitivity inversion experiments comprise 31 tests designed to provide a comprehensive evaluation of the system. To facilitate a clearer discussion of their impacts, we categorized these tests into four classes based on their roles within the system: prior information, GEOS-Chem model resolution, satellite observational constraints, and inversion system parameters (Fig. 1 and Table 2). Each test is

conducted as a controlled experiment, where only one parameter is altered while the rest remain the same as their Base inversion setting. The rationale behind the settings and their design will be elaborated in the following sections."

Sub-titles of 2.2.1: "Modifying prior emission estimates"

Sub-titles of 2.2.2: "Employing coarser model resolution"

Sub-titles of 2.2.3: "Changing satellite observational constraints"

Sub-titles of 2.2.4: "Tests on inversion system parameters"

Besides, we have added a Fig. S2 displaying MEIC inventory in SI as suggested.

[Figure]

**Figure S2. Sectoral NO$_x$ emissions from MEIC inventory in 2019 (0.25°×0.25°).**

5. From Line 114, "… while the CO$_2$ EFs are assumed to remain unchanged". If the emissions of NO$_2$, NO$_x$, and CO$_2$ are linked (Lines 38-40) what is the logic behind the assumption that CO$_2$ EFs remain unchanged? Should a scaling factor not be applied as well? This is not well explained.

**Response:**

The co-emission of CO$_2$ and NO$_x$ does not imply aligned trends in their emission factors (EFs). NO$_x$ EFs have consistently declined due to targeted end-of-pipe controls, with research documenting a continuous decrease in NO$_x$ emissions in China since 2012, supporting this downward trend in NO$_x$ EFs. In contrast, CO$_2$ EFs are primarily influenced by fuel type and combustion conditions, which have remained stable over time. We have added explanations in Lines 135-139 in Manuscript and Text S1 in SI.

Lines 135-139: "The CO$_2$-to-NO$_x$ emission ratios in 2022 are updated by reducing NO$_x$ emission factors (EFs) while keeping CO$_2$ EFs unchanged based on 2019 MEIC. The default assumption that the reduction rate halves annually is due to the limited potential for further reductions. In contrast, the CO$_2$ EFs are assumed to remain unchanged, as

they are primarily determined by fuel type and combustion conditions (Cheng et al., 2021) (details seen in Text S2)."

**Text S1. Bottom-up estimates**

To derive a sector-specific prior, we update the 2019 Multi-resolution Emission Inventory for China (MEIC) (Zheng et al., 2018) using a range of activity data. The bottom-up estimation follows two primary steps: first, we apply monthly updates based on year-on-year national activity ratios obtained from the National Bureau of Statistics (https://data.stats.gov.cn/english/easyquery.htm?cn=C01); second, we disaggregate monthly emissions into daily estimates using multi-source data. The specific data sources used in this bottom-up approach are detailed in Table S1.

For emission factors (EFs), we assume a yearly halving of the reduction rate in $NO_x$ EFs. Since 2012, $NO_x$ emissions have sharply decreased due to effective pollution control measures with many end-of-pipe devices; however, the rate of decline has slowed in recent years, reflecting the diminishing potential for further reductions (Geng et al., 2024; Li et al., 2023). As such, the default assumption is that the reduction rate in $NO_x$ EFs halves each year, consistent with the limited potential for continued reductions. By contrast, $CO_2$ EFs are assumed to remain constant over time, as they are primarily influenced by fuel type and combustion conditions (Cheng et al., 2021).

6. In Lines 88-89: "assuming that each grid's emission variability was primarily driven by its dominant source sectors (contributing over 50%)…". What about situations where no sectors make up more than 50% of a grid cell? Hypothetically, what if Power, Industry, Residential, and Transport all made up 25% of a grid cell? Do these situations not exist in the prior emission inventory? If not, why not? How is an observation-driven posterior estimate assigned to a grid cell when it doesn't meet the criteria?

**Response:**

For grids without a sector contributing over 50%, we excluded them from sectoral scaling factor calculations, instead applying scaling factors derived from grids meeting this criterion. Notably, over 80% of the grids have a sector contributing more than 50%, indicating a clear dominant sector for the majority of grids.

The overall $NO_x$ emissions remain unaffected by this threshold parameter, as they are determined prior to disaggregation into sectors (Eq. 1). The threshold mainly impacts the sectoral distribution and the $CO_2$ emissions conversion process. We assessed the threshold's effect by adjusting it to 40% and 60% (thre_40% and thre_60%), and the results show that only residential emissions exhibit sensitivity due to their relatively low share of total emissions (Fig. 4 and Fig. S13).

We have added this explanation in Lines 123-126 in Manuscript.

Lines 123-126: "For grids without a sector contributing over 50%, we excluded them from sectoral scaling factor calculations, instead applying scaling factors derived from grids meeting this criterion. The number of these grids accounts for less than 20% of total grids, making their impact negligible."

**Minor Comment**

7. For readers who are not familiar with the mass balance inversion method, providing an additional citation, or explicitly pointing the reader to an additional resource, would be more helpful than simply citing Zheng et al., 2020 and Li et al., 2023. Pointing the readers to a paper such as Mun et al., 2023 or something similar will help make the connection between the inversion system being discussed and the corresponding equations 1-4.

(Mun: https://www.sciencedirect.com/science/article/pii/S1352231022004940)

**Response:**

We have added some introduction to the mass balance method in Lines 48-50 in the Introdution.

Lines 48-50: "This short lifespan of $NO_2$ facilitates mass-balance approaches for estimating $NO_x$ emissions, which rely on the assumption of a linear relationship between $NO_2$ columns and local $NO_x$ emissions (Cooper et al., 2017; Mun et al., 2023; Martin et al., 2003)."

8. Remove the word "here" in Line 59.

**Response:**

We have removed "here" in Line 69 (original 59) as suggested.

9. Add "of" in Line 77. "ten-day moving average of anthropogenic $NO_x$ and $CO_2$"

**Response:**

We have added "of" in Line 89 (original 77) as suggested.

10. I understand the need to be succinct in Lines 78-81 regarding the scaling of emission sectors; however, it is my opinion that a little more information should be included here. The authors should consider including an extra statement explaining where these indicators came from. Were they from external an external inventory? Where they part of MEIC? Does MEIC contain sector-specific information already?

**Response:**

We have added more details regarding the bottom-up estimates in Text S1, along with Table S1 in SI, which outlines the data sources for activity levels.

**Text S1. Bottom-up estimates**

To derive a sector-specific prior, we update the 2019 Multi-resolution Emission Inventory for China (MEIC) (Zheng et al., 2018) using a range of activity data. The bottom-up estimation follows two primary steps: first, we apply monthly updates based on year-on-year national activity ratios obtained from the National Bureau of Statistics (https://data.stats.gov.cn/english/easyquery.htm?cn=C01); second, we disaggregate monthly emissions into daily estimates using multi-source data. The specific data sources used in this bottom-up approach are detailed in Table S1.

For emission factors (EFs), we assume a yearly halving of the reduction rate in $NO_x$ EFs. Since 2012, $NO_x$ emissions have sharply decreased due to effective pollution control measures with many end-of-pipe devices; however, the rate of decline has slowed in recent years, reflecting the diminishing potential for further reductions (Geng et al., 2024; Li et al., 2023). As such, the default assumption is that the reduction rate in $NO_x$ EFs halves each year, consistent with the limited potential for continued reductions. By contrast, $CO_2$ EFs are assumed to remain constant over time, as they are primarily influenced by fuel type and combustion conditions (Cheng et al., 2021).

**Table S1. Data sources used in the bottom-up estimates.**

| Steps | Corresponding MEIC sector | Adopted data | Data source |
|---|---|---|---|
| Monthly emission estimation* | Power | Thermal power generation | National Bureau of Statistics (https://data.stats.gov.cn/english/easyquery.htm?cn=C01) |
| | Cement | Cement production | |
| | Iron | Iron production | |
| | Other industry | Manufacturing value added | |
| | On-road | Road Freight turnover | |
| | Off-road | Construction area | |
| Dissolving monthly emissions into daily | Residential/ Residential-bio | Population-weighted heating degree day | Calculation based on the 2m temperature data from the ERA5 dataset |
| | Power/ Cement/ Other industry | Coal consumption | (Wu et al., 2022) |
| | Iron | Operating rates of electric furnace | The custeel database (https://www.custeel.com/) |
| | On-road/ Off-road | Baidu migration data | The Baidu database (https://qianxi.baidu.com/) |

*Production index are used to differentiate January and February from the combined first two months' data in the National Bureau of Statistics.

11. The source of the 40% reduction is confusing (Lines 105-106). Only after reading the rest of the paper did I realize that this was from one of the sensitivity tests. (Again, the authors need to focus on the logical flow of information in the text.)

**Response:**

The 40% reduction in simulation is used to quantify the response of $NO_2$ concentration to the changes in anthropogenic $NO_x$ emission ($\beta$), building on previous works. In our previous tests, this perturbation magnitude seems to have a limited impact on final estimates within the tested range of 30-50%. We have added a brief explanation in Lines 110-112 in Manuscript. Besides, we have made adjustments in Methods to clarify the logical flow, please refer to the response to Comment 4.

Lines 110-112: "The 40% reduction was selected after a series of sensitivity tests, which

demonstrated that this perturbation level exerts a limited impact on the $\beta$ estimates (Zheng et al., 2020)."

12. Section 2.2.1 does not mention the spatial resolution of the inventory.

**Response:**

The original MEIC inventory has a resolution of 0.25°×0.25°, which we aggregate to 0.5°×0.625° to align with the resolution of the prior and the GEOS-Chem model. We have added this explanation in Lines 94-96 and Lines 165-166.

Lines 94-96: "Notably, to reconcile the resolution between the prior emissions and the model, we aggregated the original MEIC emissions from a resolution of 0.25°×0.25° (Fig. S2) to 0.5°×0.625°."

Lines 165-166: "The prior provides the sectoral profile for subsequent emission attribution. We conducted a comprehensive examination of associated parameters when updating the prior from 2019 MEIC (0.5°×0.625°),"

13. In Line 172, consider changing "policies" to "protocols". The use of "policies" has political connotations.

**Response:**

We have changed the "policies" to "protocols" in Line 206 (original 172) as suggested.

14. In Line 245, add "the" before "tests' impact".

**Response:**

We have added "the" in Line 278 (original 245) as suggested.

15. From Line 252, "A reduction in $NO_x$ increases $rNO_x$". Why is this the case? I do not follow.

**Response:**

$rNO_x$ represents the reduction ratio of $NO_x$ emission factors (EFs); thus, a greater reduction in $NO_x$ EFs corresponds to a higher $rNO_x$ value. We have explained this parameter in Line 143.

Line 143: "$rNO_{x\ s,i,y}$ is the reduction ratio in $NO_x$ EFs by sector from 2019 to 2022 derived from the bottom-up estimation."

16. In Line 273, I think "parameters" should be singular: "parameter".

**Response:**

We have corrected the "parameters" to "parameter" in Line 306 (original 273) as suggested.

17. In Line 307, "mode" should be "model".

300  **Response:**

We have changed the "mode" to "model" in Line 340 (original 307) as suggested.

18. How are the cities arranged in Figure 5? Are they arranged by longitude?

**Response:**

305  The original arrangement was based on the IDs of China's provinces. We have now modified it to follow an area-based sequence, as the area is one of the key factors influencing regional emission estimates in this methodology.

[Figure]

**Figure 5**. **Response of provincial annual total NO$_x$ and CO$_2$ emissions to different**
310  **tests.** (**a**) and (**b**) show $RC_p$ of NO$_x$ emissions incurred by tests. (**c**) and (**d**) are plotted for CO$_2$ emission as (**a**) and (**b**). Lines refer to the $RC_p$ caused by the corresponding test or the averaged $RC_p$ caused by corresponding test clusters (ef_[-10%, -1%] and β_[-20, 20%]), and the shadow refers to the $RC_p$ range in test clusters. Only provinces with enough TROPOMI observations are shown here (i.e., grids with NO$_2$ TVCDs larger
315  than $1 \times 10^{15}$ molecules/cm$^2$ cover more than 90% of anthropogenic NO$_x$ emissions within provinces). The provinces are arranged by area.

**Reference:**

320    Cheng, J., Tong, D., Liu, Y., Bo, Y., Zheng, B., Geng, G., He, K., and Zhang, Q.: Air quality and health benefits of China's current and upcoming clean air policies, Faraday Discussions, 226, 584-606, https://doi.org/10.1039/D0FD00090F, 2021.

Cooper, M., Martin, R. V., Padmanabhan, A., and Henze, D. K.: Comparing mass balance and adjoint methods for inverse modeling of nitrogen dioxide columns for global nitrogen oxide emissions, Journal

325    of Geophysical Research: Atmospheres, 122, 4718-4734, https://doi.org/10.1002/2016JD025985, 2017.

Cooper, M. J., Martin, R. V., Hammer, M. S., Levelt, P. F., Veefkind, P., Lamsal, L. N., Krotkov, N. A., Brook, J. R., and McLinden, C. A.: Global fine-scale changes in ambient $NO_2$ during COVID-19 lockdowns, Nature, 601, 380-387, https://doi.org/10.1038/s41586-021-04229-0, 2022.

Geng, G., Liu, Y., Liu, Y., Liu, S., Cheng, J., Yan, L., Wu, N., Hu, H., Tong, D., Zheng, B., Yin, Z., He,

330    K., and Zhang, Q.: Efficacy of China's clean air actions to tackle $PM_{2.5}$ pollution between 2013 and 2020, Nature Geoscience, 17, 987-994, 10.1038/s41561-024-01540-z, 2024.

Li, H. and Zheng, B.: Toward monitoring daily anthropogenic $CO_2$ emissions with air pollution sensors from space, One Earth, 7, 1846-1857, 10.1016/j.oneear.2024.08.019, 2024.

Li, S., Wang, S., Wu, Q., Zhang, Y., Ouyang, D., Zheng, H., Han, L., Qiu, X., Wen, Y., Liu, M., Jiang, Y.,

335    Yin, D., Liu, K., Zhao, B., Zhang, S., Wu, Y., and Hao, J.: Emission trends of air pollutants and $CO_2$ in China from 2005 to 2021, Earth Syst. Sci. Data, 15, 2279-2294, https://doi.org/10.5194/essd-15-2279-2023, 2023.

MacDonald, C. G., Mastrogiacomo, J. P., Laughner, J. L., Hedelius, J. K., Nassar, R., and Wunch, D.: Estimating enhancement ratios of nitrogen dioxide, carbon monoxide and carbon dioxide using satellite

340    observations, Atmos. Chem. Phys., 23, 3493-3516, https://doi.org/10.5194/acp-23-3493-2023, 2023.

Martin, R. V., Jacob, D. J., Chance, K., Kurosu, T. P., Palmer, P. I., and Evans, M. J.: Global inventory of nitrogen oxide emissions constrained by space-based observations of $NO_2$ columns, Journal of Geophysical Research: Atmospheres, 108, https://doi.org/10.1029/2003JD003453, 2003.

Mun, J., Choi, Y., Jeon, W., Lee, H. W., Kim, C.-H., Park, S.-Y., Bak, J., Jung, J., Oh, I., Park, J., and

345    Kim, D.: Assessing mass balance-based inverse modeling methods via a pseudo-observation test to constrain $NO_x$ emissions over South Korea, Atmospheric Environment, 292, 119429, https://doi.org/10.1016/j.atmosenv.2022.119429, 2023.

Roten, D., Lin, J. C., Das, S., and Kort, E. A.: Constraining Sector-Specific $CO_2$ Fluxes Using Space-Based $XCO_2$ Observations Over the Los Angeles Basin, Geophysical Research Letters, 50,

350    e2023GL104376, https://doi.org/10.1029/2023GL104376, 2023.

Taylor, T. E., O'Dell, C. W., Baker, D., Bruegge, C., Chang, A., Chapsky, L., Chatterjee, A., Cheng, C., Chevallier, F., Crisp, D., Dang, L., Drouin, B., Eldering, A., Feng, L., Fisher, B., Fu, D., Gunson, M., Haemmerle, V., Keller, G. R., Kiel, M., Kuai, L., Kurosu, T., Lambert, A., Laughner, J., Lee, R., Liu, J., Mandrake, L., Marchetti, Y., McGarragh, G., Merrelli, A., Nelson, R. R., Osterman, G., Oyafuso, F.,

355    Palmer, P. I., Payne, V. H., Rosenberg, R., Somkuti, P., Spiers, G., To, C., Weir, B., Wennberg, P. O., Yu, S., and Zong, J.: Evaluating the consistency between OCO-2 and OCO-3 $XCO_2$ estimates derived from the NASA ACOS version 10 retrieval algorithm, Atmos. Meas. Tech., 16, 3173-3209, 10.5194/amt-16-3173-2023, 2023.

Wu, N., Geng, G., Qin, X., Tong, D., Zheng, Y., Lei, Y., and Zhang, Q.: Daily Emission Patterns of Coal-

360    Fired Power Plants in China Based on Multisource Data Fusion, ACS Environmental Au, https://doi.org/10.1021/acsenvironau.2c00014, 2022.

Yang, E. G., Kort, E. A., Wu, D., Lin, J. C., Oda, T., Ye, X., and Lauvaux, T.: Using Space-Based

Observations and Lagrangian Modeling to Evaluate Urban Carbon Dioxide Emissions in the Middle East, Journal of Geophysical Research: Atmospheres, 125, e2019JD031922, https://doi.org/10.1029/2019JD031922, 2020.

Ye, C., Zhou, X., Pu, D., Stutz, J., Festa, J., Spolaor, M., Tsai, C., Cantrell, C., Mauldin, R. L., Campos, T., Weinheimer, A., Hornbrook, R. S., Apel, E. C., Guenther, A., Kaser, L., Yuan, B., Karl, T., Haggerty, J., Hall, S., Ullmann, K., Smith, J. N., Ortega, J., and Knote, C.: Rapid cycling of reactive nitrogen in the marine boundary layer, Nature, 532, 489-491, 10.1038/nature17195, 2016.

Ye, X., Lauvaux, T., Kort, E. A., Oda, T., Feng, S., Lin, J. C., Yang, E. G., and Wu, D.: Constraining Fossil Fuel $CO_2$ Emissions From Urban Area Using OCO-2 Observations of Total Column $CO_2$, Journal of Geophysical Research: Atmospheres, 125, e2019JD030528, https://doi.org/10.1029/2019JD030528, 2020.

Zheng, B., Geng, G., Ciais, P., Davis, S. J., Martin, R. V., Meng, J., Wu, N., Chevallier, F., Broquet, G., Boersma, F., van der A, R., Lin, J., Guan, D., Lei, Y., He, K., and Zhang, Q.: Satellite-based estimates of decline and rebound in China's $CO_2$ emissions during COVID-19 pandemic, Science Advances, 6, eabd4998, https://doi.org/10.1126/sciadv.abd4998, 2020.

Zheng, B., Tong, D., Li, M., Liu, F., Hong, C., Geng, G., Li, H., Li, X., Peng, L., Qi, J., Yan, L., Zhang, Y., Zhao, H., Zheng, Y., He, K., and Zhang, Q.: Trends in China's anthropogenic emissions since 2010 as the consequence of clean air actions, Atmos. Chem. Phys., 18, 14095-14111, 10.5194/acp-18-14095-2018, 2018.